# MDTree: A Masked Dynamic Autoregressive Model for Phylogenetic Inference

**Zelin Zang**[*]  *zangzelin@westlake.edu.cn*
*Centre for Artificial Intelligence and Robotics (CAIR), Hong Kong Institute of Science and Innovation (HKISI)*
*School of Engineering, Westlake University*

**Chenrui Duan**[*]  *chenruiduan@westlake.edu.cn*
*School of Engineering, Westlake University*

**Siyuan Li**  *siyuanli@westlake.edu.cn*
*School of Engineering, Westlake University*

**Jinlin Wu**  *jinlinwu@hkisi-cas.org.hk*
*Centre for Artificial Intelligence and Robotics (CAIR), Hong Kong Institute of Science and Innovation (HKISI)*

**BingoWing-Kuen Ling**  *wkling@tias.ac.cn*
*Center for the Integrated Circuits and Artificial Intelligence, Tsientang Institute for Advanced Study, Zhejiang 310024*

**Fuji Yang**  *fjyang@tias.ac.cn*
*Center for the Integrated Circuits and Artificial Intelligence, Tsientang Institute for Advanced Study*

**Jiebo Luo**  *jluo@hkisi-cas.org.hk*
*Hong Kong Institute of Science and Innovation (HKISI)*

**Zhen Lei**[†]  *zhen.lei@ia.ac.cn*
*Centre for Artificial Intelligence and Robotics (CAIR), Hong Kong Institute of Science and Innovation (HKISI)*
*Institute of Automation, Chinese Academy of Sciences (CASIA)*
*School of Artificial Intelligence, University of Chinese Academy of Sciences (UCAS)*

**Stan Z. Li**[†]  *stan.z.li@westlake.edu.cn*
*School of Engineering, Westlake University, Hangzhou, Zhejiang 310030, China*

**Reviewed on OpenReview:** *https://openreview.net/forum?id=dTSptQNygv*

## Abstract

Phylogenetic tree inference requires optimizing both branch lengths and topologies, yet traditional MCMC-based methods suffer from slow convergence and high computational cost. Recent deep learning approaches improve scalability but remain constrained: Bayesian models are computationally intensive, autoregressive methods depend on fixed species orders, and flow-based models underutilize genomic signals. Fixed-order autoregression introduces an inductive bias misaligned with evolutionary proximity: early misplacements distort subsequent attachment probabilities and compound topology errors (exposure bias). Absent sequence-informed priors, the posterior over the super-exponential topology space remains diffuse and multimodal, yielding high-variance gradients and sluggish convergence for both MCMC proposals and neural samplers. We propose MDTree, a masked dynamic autoregressive framework that integrates genomic priors into a Dynamic Ordering Network to learn biologically informed node sequences. A dynamic masking mechanism further enables parallel node insertion, improving efficiency without sacrificing accuracy. Experiments on

---

[*]Both authors contributed equally to this research.
[†]Corresponding author.

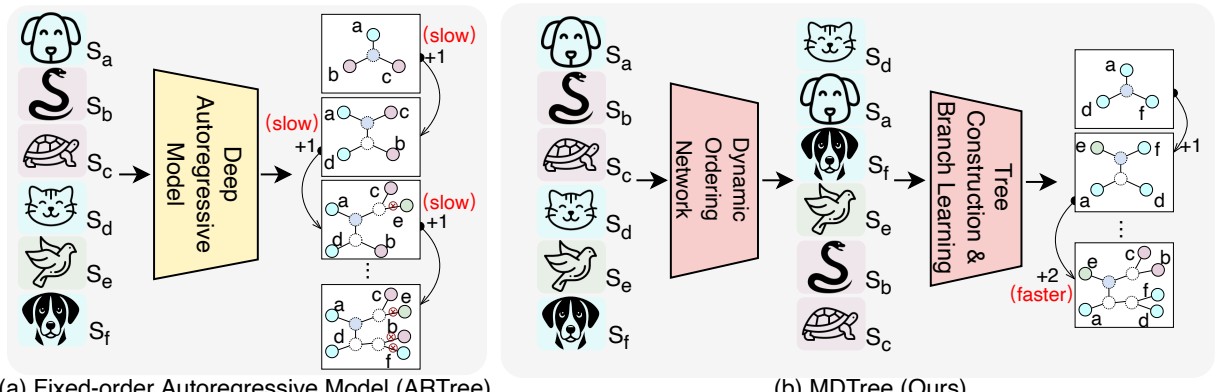

Figure 1: **Comparison between fixed-order autoregressive generation (ARTree) and our dynamic-order method (MDTree).** (a) Fixed-order ARTree adds species sequentially according to a predefined order, ignoring biological priors. This leads to suboptimal intermediate structures and slower generation, as only one leaf node is added at each step. (b) MDTree (ours) employs a Dynamic Ordering Network to determine a biologically-informed insertion order based on genomic features, enabling related species to be clustered earlier. The Tree Construction & Branch Learning module further supports parallel insertion of multiple nodes, achieving faster generation and more phylogenetically consistent topologies.

standard benchmarks demonstrate that MDTree outperforms existing methods in accuracy and runtime while producing biologically coherent phylogenies, providing a scalable solution for large-scale evolutionary analysis.

# 1 Introduction

Phylogenetic trees are fundamental for revealing evolutionary relationships, enabling lineage tracing from common ancestors to present-day organisms using DNA or protein sequences (Brocchieri, 2001; Munjal et al., 2019). They underpin studies in taxonomy, evolutionary biology, and medicine, offering insights into species origins, biodiversity, and the evolutionary trajectories of pathogens and cancer cells (Hugenholtz et al., 2021; Tummers & Green, 2022). Accurate and efficient inference has high practical value: in pathogen source tracing, it supports timely outbreak interventions (Biek et al., 2015); in cancer evolution analysis (Fimereli et al., 2022), it reveals clonal architecture and treatment resistance; and in biodiversity conservation (Theissinger et al., 2023), it enables large-scale, automated species relationship reconstruction. These applications underscore both the scientific and societal significance of phylogenetic modeling. Yet, the surge of genomic data and the combinatorial growth of tree topologies pose major computational challenges, calling for scalable and accurate new methods.

Traditional statistical frameworks, notably Maximum Likelihood Estimation (MLE)(Izquierdo-Carrasco et al., 2011; Solís-Lemus & Ané, 2016) and Bayesian Inference (BI) via Markov Chain Monte Carlo (MCMC)(Zhang et al., 2018; Wang et al., 2020), have long underpinned phylogenetic inference. Yet, with increasing taxa, they encounter severe computational bottlenecks: the space of unrooted bifurcating topologies grows super-exponentially as $(2N-5)!!$, while the joint optimization of continuous branch lengths and discrete topologies further compounds complexity.

Leveraging deep learning, breakthroughs in phylogenetic inference have burst onto the scene, addressing long-standing computational challenges in the field (Nesterenko et al., 2022; Smith & Hahn, 2023; Tang et al., 2024). Research efforts primarily follow two main directions: representation learning on known tree structures and generative models. The former, exemplified by VBPI-GNN (Zhang, 2023), optimizes performance based on predefined topologies but struggles when the topology is unknown and both topology and branch lengths must be inferred. These methods also underutilize evolutionary information from biological sequences, impacting accuracy and flexibility (Penny, 2004). On the other hand, generative models, which

infer tree structures directly from data, can be further divided into three types: Bayesian generative models (e.g., Geophy (Mimori & Hamada, 2024)) leverage probabilistic frameworks to capture uncertainty but are computationally intensive; autoregressive models (e.g., ARTree (Xie & Zhang, 2024)) sequentially add nodes, offering flexibility yet relying on predefined orders that overlook true evolutionary relationships, while their stepwise nature leads to inefficiency for large datasets (Razavi et al., 2019). Lastly, Generative Flow Networks (GFNs) (e.g., PhyloGFN (Zhou et al., 2024)) provide greater flexibility by exploring multimodal posterior distributions but still struggle to fully integrate evolutionary signals, impacting the accuracy of inferred trees. Therefore, few methods have achieved these goals simultaneously.

To overcome these limitations, we focus on a core question: *how can biological priors effectively guide node addition to improve phylogenetic inference accuracy?* As shown in Fig. 2, classical autoregressive methods (Fig. 2a) rely on fixed orders (e.g., lexicographical), overlooking evolutionary relationships and often producing inaccurate trees 1. Our method (Fig. 2b) learns evolutionarily meaningful node orders, ensuring species like reptiles, birds, and mammals are added in line with their ancestry. This improves the accuracy and biological relevance of generated trees by prioritizing species with closer common ancestors.

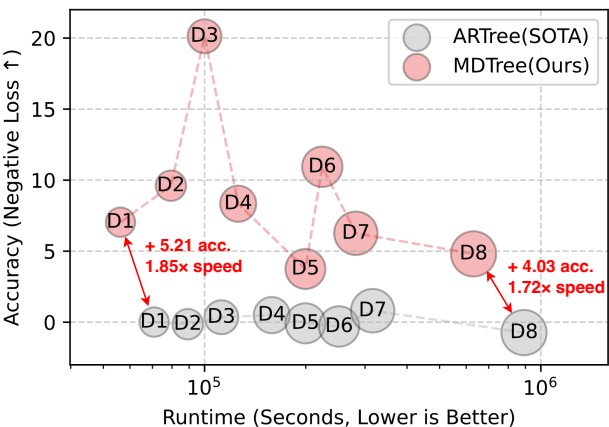

Figure 2: **Runtime and node count comparison between MDTree and ARTree.** Evaluation is conducted on eight benchmarks under two optimization settings (log-scale).

Specifically, we adopt a **dynamic autoregressive generation paradigm**, where both the order of node additions and their insertion positions are learned from genomic sequences instead of being fixed in advance. This paradigm is instantiated by our *Masked Dynamic Autoregressive Model* (MDTree), which integrates a Diffusion Ordering Network (DON) to learn biologically informed orders directly from sequence data via an absorbing diffusion model (Bond-Taylor et al., 2021), mitigating the limitations of fixed or random orders. By combining the strengths of Graph Neural Networks and Language Models (LMs), MDTree captures intricate genomic relationships while modeling complex tree structures. A Dynamic Masking Mechanism enables parallel node processing, improving efficiency. Lastly, we employ a dual-pass tree traversal strategy for branch length estimation and use the LAX model (Grathwohl et al., 2017) to reduce variance in discrete sampling for stabilizing optimization and enhancing convergence.

Experiments on phylogenetic benchmarks show that MDTree outperforms existing methods in accuracy and efficiency. Empirical analysis of Angiosperms353 (Zuntini et al., 2024) further demonstrates its ability to recover evolutionary lineages, including Rosaceae and Moraceae, suggesting broader biological applications. In summary, our contributions are:

- **A novel dynamic autoregressive generation paradigm for phylogenetic inference**: We leverage a generation strategy that dynamically learns node order and insertion positions from genomic sequence data, improving the accuracy and biological relevance of inferred trees.

- **An innovative methodology**: We propose MDTree, which integrates a Diffusion Ordering Network for biologically informed node orders, combines genomic Language Models with dual-pass traversal for precise tree generation, and employs a dynamic masking mechanism for efficient parallel processing.

- **Strong experimental validation**: Comprehensive experiments validate that MDTree achieves state-of-the-art performance. Visualizations from real-world Angiosperm datasets further confirm the biological relevance and interpretability of the generated trees.

## 2 Related Works

Phylogenetic inference methods are generally categorized into traditional and deep learning-based approaches; each is further divided into graph structure generation and representation models. For details on background, please see Appendix A.

**Traditional Methods** rely on predefined evolutionary models and statistical inference. *Graph Structure Generation Models:* MrBayes (Ronquist et al., 2012) utilizes Bayesian inference to generate trees but struggles with high-dimensional combinatorial spaces, requiring large sample sizes for accuracy. VaiPhy (Koptagel et al., 2022) combines SLANTIS sampling strategy (Diaconis, 2019) with biological models (e.g., JC model (Munro, 2012)) to estimate branch lengths and generate accurate tree structures. *Graph Structure Representation Models:* SBN (Zhang & Matsen IV, 2018a) models the probability distribution of tree topologies from existing trees, focusing on subsplit relationships without directly estimating branch lengths. VBPI (Zhang & Matsen IV, 2018b) extends SBNs to estimate posterior distributions and optimize branch lengths through variational inference.

**Deep Learning-based Methods** offer more flexible and scalable solutions. *Graph Structure Generation Models:* (1) Bayesian Generative Models like GeoPhy (Mimori & Hamada, 2024) learn latent tree representations to generate diverse topologies. (2) Autoregressive Models such as ARTree (Xie & Zhang, 2024) sequentially generate trees, well-suited for hierarchical data. (3) Generative Flow Networks like PhyloGFN (Zhou et al., 2024) optimize tree generation paths using Markov decision processes. *Graph Structure Representation Models:* VBPI-GNN (Zhang, 2023) combines SBNs with variational inference to optimize topology and branch lengths.

## 3 Background

### 3.1 Phylogeny and Machine Learning Applications in Biology

Phylogeny is the study of evolutionary relationships among species, aiming to infer their common ancestors and evolutionary paths by analyzing gene or protein sequences. Phylogenetic trees are widely used in biology to represent these relationships, providing insights into species origins, biodiversity, and evolutionary trajectories. Phylogenetic trees play a crucial role in various applications, including pathogen source tracing, cancer evolution analysis, and biodiversity conservation. However, with the surge in genomic data, phylogenetic inference faces substantial computational challenges, especially when inferring trees for a large number of species. Traditional methods encounter significant computational bottlenecks as the number of taxa increases, requiring more time and computational resources.

**Challenges in Traditional Phylogenetic Inference Methods.** Traditional phylogenetic inference methods, such as Maximum Likelihood Estimation (MLE) and Bayesian Inference (MCMC), rely on exhaustive searches over large tree spaces to calculate the optimal topology. As the number of species increases, the combinatorial space of possible tree topologies grows exponentially, leading to severe computational bottlenecks. For example, Bayesian inference methods are computationally expensive and slow, particularly as the number of taxa increases. Additionally, traditional autoregressive models (e.g., ARTree) rely on predefined species orders, which often do not align with the actual evolutionary relationships, resulting in suboptimal tree structures and slower convergence.

**The Role of Machine Learning in Phylogenetic Inference.** In recent years, deep learning methods have shown great potential in improving phylogenetic inference by leveraging complex relationships in genomic data. These methods allow for adaptive learning of node orders, which helps overcome the limitations of fixed species orders in traditional models. For instance, Generative Flow Networks (GFNs) and autoregressive models, like ARTree, have improved the efficiency and accuracy of tree generation. However, these methods still fail to fully incorporate biological prior knowledge, such as evolutionary relationships between species, and often do not capture the complete genomic signals, limiting their ability to provide accurate phylogenies for large datasets.

**Innovations of MDTree.** To address the limitations of traditional methods, we propose MDTree, a novel dynamic autoregressive framework based on a Dynamic Ordering Network (DON). MDTree dynamically learns the node addition order from genomic sequence data, instead of relying on predefined orders. This approach ensures that the tree construction process better reflects the true evolutionary relationships among species, improving both accuracy and biological relevance. Furthermore, MDTree incorporates a dynamic masking mechanism that enables parallel insertion of nodes, significantly improving computational efficiency. By leveraging this method, we not only overcome computational bottlenecks but also ensure that the generated phylogenetic trees are biologically consistent, making them suitable for large-scale evolutionary analysis.

## 3.2 Phylogenetic Posterior and Variational Inference (VI)

Variational Autoencoders (VAE) Kingma & Welling (2013) are deep generative models that learn the input data distribution by encoding it into a latent space. In this process, the encoder maps each input $x$ to a latent space defined by parameters: mean $\mu$ and variance $\sigma$. Latent variables $z$ are then sampled from this distribution for data generation.

Variational Inference (VI) is employed within VAEs to handle the computational challenges of estimating marginal likelihoods of observed data. VI approximates the marginal likelihood using a variational distribution $q_\phi(z|x)$ to estimate the posterior. The goal of VI is to maximize the Evidence Lower Bound (ELBO), formulated as:

$$\text{ELBO} = \mathbb{E}_{q_\phi(z|x)}[\log p_\theta(x|z)] - \text{KL}[q_\phi(z|x)||p(z)] \tag{1}$$

The first term is the reconstruction log-likelihood, $\log p_\theta(x|z)$, which can be considered as a decoder, i.e., the log-likelihood between the reconstructed data and the original data given the potential representation. The second term, the Kullback-Leibler (KL) divergence, quantifies the difference between the variational posterior $q_\phi(z|x)$ and the latent prior $p(z)$.

In the context of phylogenetic inference, VI helps to approximate the posterior distribution of tree topologies and branch lengths, which are often intractable to compute directly. By applying VAE with VI, we can efficiently infer phylogenetic structures while maintaining the biological relevance of the tree, improving both the accuracy and computational efficiency of the process.

# 4 Methods

## 4.1 Problem Formulation and Notation

A unified model that can handle both tasks must therefore (i) capture biologically meaningful topological structures, and (ii) accurately estimate continuous evolutionary distances, while being robust to limited or no supervision on the topology. Unless otherwise specified, all vector representations (e.g., $\mathbf{h}_i$) are treated as column vectors.

**Formulation.** Given a set of $N$ species sequences $\mathcal{S} = \{s_i\}_{i=1}^N$ and their multiple sequence alignment (MSA) $\mathcal{A}$, along with genomic representations $\mathcal{G} = \{\mathbf{g}_i\}_{i=1}^N$ extracted from a pretrained Genomic Language Model (e.g., DNABERT2 (Zhou et al., 2023)), we aim to infer a phylogenetic tree that captures both its discrete topology and continuous evolutionary distances. Formally, the tree is modeled as an unrooted binary graph $\tau = (\mathcal{V}_\tau, \mathcal{E}_\tau)$, where $\mathcal{V}_\tau$ denotes the set of taxa and internal nodes, and $\mathcal{E}_\tau$ the set of undirected edges between them. Each edge $e \in \mathcal{E}_\tau$ is associated with a branch length $b_e \in \mathbb{R}_+$, and we denote the vector of all branch lengths as $\boldsymbol{\ell}_\tau = (b_e : e \in \mathcal{E}_\tau) \in \mathbb{R}_+^{|\mathcal{E}_\tau|}$. Our objective is to learn a mapping, $\mathcal{F} : (\mathcal{S}, \mathcal{A}, \mathcal{G}) \longrightarrow (\tau, \boldsymbol{\ell}_\tau)$, which jointly specifies the topology $\tau$ and its associated branch lengths $\boldsymbol{\ell}_\tau$.

**Task Definitions.** We evaluate MDTree on two complementary phylogenetic inference tasks:

*Tree Topology Density Estimation (TDE).* This task focuses on learning a flexible distribution $q_\theta(\tau)$ over tree topologies that captures the uncertainty inherent in phylogenetic inference. The model is trained to maximize the marginal log-likelihood:

$$\mathbb{E}_{q_\theta(\tau)}[\log p(\mathcal{A} \mid \tau)], \quad \text{where} \quad p(\mathcal{A} \mid \tau) = \int p(\mathcal{A} \mid \tau, \boldsymbol{\ell}_\tau) p(\boldsymbol{\ell}_\tau \mid \tau) \, d\boldsymbol{\ell}_\tau. \tag{2}$$

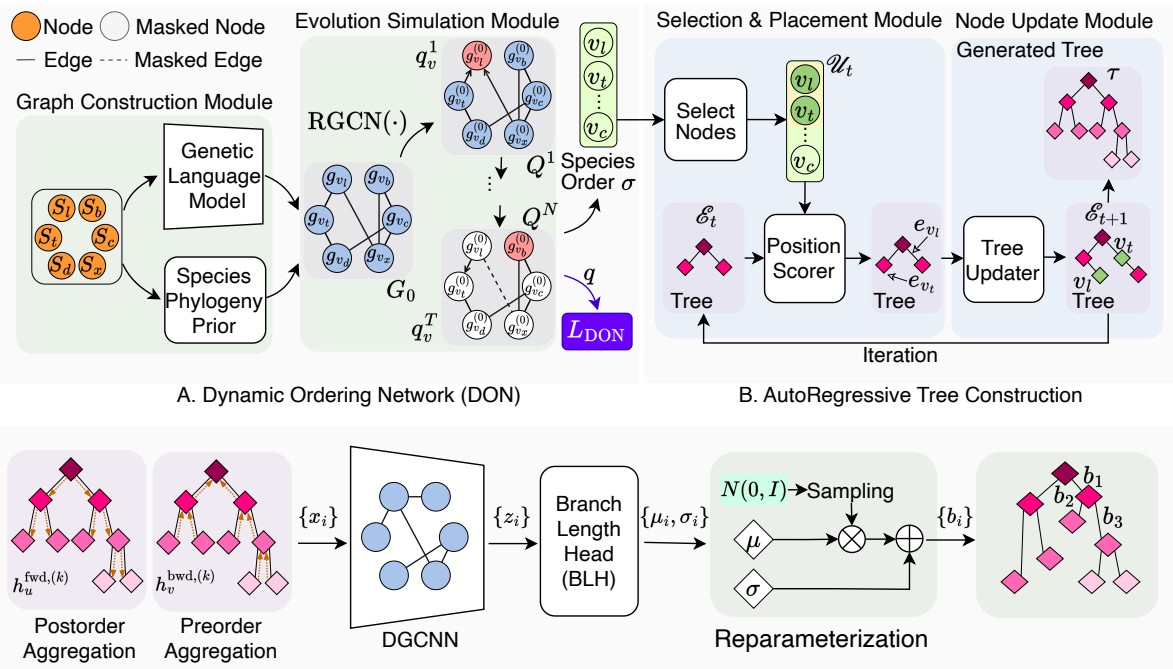

Figure 3: **Framework of MDTree for dynamic autoregressive tree generation.** A. The Dynamic Ordering Network module utilizes a pre-trained genomic LM to extract embeddings from sequences $\mathcal{S}$, guiding nodes into absorbing states in an autoregressive manner as determined by DON $q_\theta(\sigma \mid \mathcal{G}_0)$. B. The Autoregressive Tree Construction module employs a parallel strategy to add multiple leaf and internal nodes simultaneously at specified positions based on the order provided by DON. C. The Branch Length Learning module optimizes branch lengths through a dual-pass traversal.

In practice, we approximate the integral using a point estimate $\hat{\ell}_\tau$ obtained from our branch length estimation module (Sec. 4.4), yielding $p(\mathcal{A} \mid \tau) \approx p(\mathcal{A} \mid \tau, \hat{\ell}_\tau)$. During TDE evaluation, we focus solely on topological accuracy metrics (e.g., Robinson-Foulds distance) and discard the branch length estimates.

*Variational Bayesian Phylogenetic Inference (VBPI).* This task extends TDE by jointly modeling both topology and branch lengths through a structured posterior $q_{\theta,\phi}(\tau, \ell_\tau) = q_\theta(\tau)q_\phi(\ell_\tau \mid \tau)$. The model optimizes the Evidence Lower Bound (ELBO):

$$\mathbb{E}_{q_\theta(\tau)q_\phi(\ell_\tau|\tau)} \left[\log p(\mathcal{A} \mid \tau, \ell_\tau)\right] - \mathrm{KL}(q_{\theta,\phi}(\tau, \ell_\tau) \, \| \, p(\tau, \ell_\tau)), \tag{3}$$

where $p(\tau, \ell_\tau) = p(\tau)p(\ell_\tau \mid \tau)$ with uniform topology prior and exponential branch length prior. During VBPI evaluation, we assess both topological accuracy and branch length estimation quality (e.g., using Euclidean distance in branch length space).

**Model Architecture.** To address the limitations of fixed node orders in prior autoregressive models (Xie & Zhang, 2024), we propose **MDTree** (Fig. 3), which dynamically learns biologically informed node addition orders and insertion positions from $\mathcal{G}$ via a Dynamic Ordering Network (DON) based on an absorbing diffusion process (Austin et al., 2021). The learned order and contextual node embeddings jointly guide an autoregressive tree construction module with dynamic masking for parallel insertion. Finally, a dual-pass traversal refines branch lengths using both global and local structural cues. MDTree supports both TDE and VBPI within a unified pipeline: the full architecture (DON + Tree Construction + Branch Length Refinement) is used for both tasks, trained with a combined objective. During evaluation, task-specific metrics are applied: for TDE, the branch length estimates $\hat{\ell}_\tau$ serve as auxiliary variables to approximate the marginal likelihood, while for VBPI, they are evaluated as part of the posterior distribution.

### 4.2 DON for Learning Biologically Informed Node Orders with Genomic Priors

The order in which species nodes are added to a phylogenetic tree significantly impacts the inferred topology, especially under the constraint of binary unrooted trees. From a biological perspective, species with closer ancestry should be introduced earlier in the tree construction process to better preserve evolutionary semantics (Penny, 2004; Gregory, 2008). While some recent works show robustness to taxa orderings (Xie & Zhang, 2024), the benefit of *learning* biologically informed node orders remains underexplored. Such an approach could allow the model to adaptively exploit genomic signals to produce topologies that better reflect true evolutionary relationships.

**Graph Construction Module**  Given an input set of sequences $\mathcal{S}$, each $s_v \in \mathcal{S}$ is first encoded into a genomic embedding $\mathbf{g}_v \in \mathbb{R}^d$ using a pretrained genomic language model (e.g., DNABERT2 (Zhou et al., 2023)), injecting biological priors into subsequent ordering decisions. An initial graph $\mathcal{G}_0 = (\mathcal{V}, \mathcal{E})$ is constructed based on sequence similarity or known homology (species phylogeny prior), which serves as the structural backbone for contextual reasoning. The initial contextual embeddings $\mathbf{h}_v^{(0)} \in \mathbb{R}^d$ for each node $v$ are computed by passing node features through a relational graph convolutional network (RGCN) (Schlichtkrull et al., 2018):

$$\mathbf{h}_v^{(0)} = \text{RGCN}\left(\mathbf{g}_v, \mathcal{E}\right), \tag{4}$$

where $\mathbf{g}_v$ is the genomic embedding of node $v$. The resulting set of embeddings $\{\mathbf{h}_v^{(0)}\}_{v \in \mathcal{V}}$ integrates both sequence-level genomic information and local graph structure information encoded in $\mathcal{E}$. These embeddings remain fixed throughout the subsequent absorbing diffusion process, which models node ordering through evolving selection probabilities rather than feature updates.

**Evolution Simulation Module**  The node ordering process models the sequential absorption of nodes into a masked state over $N$ discrete time steps $t = 1, 2, \ldots, N$. At each step $t$, the model selects one node from the remaining active set $\mathcal{V}_t$ to add to the ordered sequence $\sigma$.

*(1) Transition dynamics:* The discrete-time transition matrix $\mathbf{Q}^t \in \mathbb{R}^{(N+1)\times(N+1)}$ controls state changes at step $t$, where each element $\mathbf{Q}_{ij}^t$ represents the probability of transitioning from state $i$ to state $j$. The matrix is designed to model gradual absorption into a masked state $m = N + 1$:

$$\mathbf{Q}_{ij}^t = \begin{cases} 1 & \text{if } i = j = m \text{ (absorbing state)} \\ 1 - \beta_t & \text{if } i = j \neq m \text{ (self-loop)} \\ \beta_t & \text{if } j = m, i \neq m \text{ (absorption)} \\ 0 & \text{otherwise,} \end{cases} \tag{5}$$

where $\beta_t = \frac{1}{2}\left(1 + \cos\left(\frac{t}{N}\pi\right)\right)$ is a monotonically decreasing schedule that controls the absorption rate, starting near 1 at $t = 1$ and decreasing to 0 at $t = N$. For the base case, we define $\bar{\mathbf{Q}}^0 = \mathbf{I}_{N+1}$ (the identity matrix).

*(2) Node selection probability:* At each step $t$, the model computes the probability of absorbing each active node $v \in \mathcal{V}_t$. The active set comprises all nodes not yet absorbed: $\mathcal{V}_t = \mathcal{V} \setminus \{v_1, \ldots, v_{t-1}\}$, where during inference $v_i = v_i^*$ (greedy selections), and during training $v_i = v_i^{\text{ref}}$ (ground-truth prefix via teacher forcing). The fixed initial embedding $\mathbf{h}_v^{(0)} \in \mathbb{R}^d$ is projected into the $(N+1)$-dimensional state space via a learned projection $\mathbf{W}_{\text{proj}} \in \mathbb{R}^{(N+1)\times d}$. Since the embeddings are fixed, the projected representation $\mathbf{z}_v$ is computed once and reused at all time steps. The selection probability vector $\mathbf{q}_v^{(t)} \in \mathbb{R}^{N+1}$ for node $v$ at step $t$ is computed by normalizing the posterior:

$$\mathbf{q}_v^{(t)} = \frac{(\mathbf{Q}^{t\top}\mathbf{z}_v) \odot (\bar{\mathbf{Q}}^{t-1\top}\mathbf{z}_v)}{\mathbf{z}_v^\top \bar{\mathbf{Q}}^t \mathbf{z}_v}, \quad \text{where} \quad \mathbf{z}_v = \mathbf{W}_{\text{proj}}\mathbf{h}_v^{(0)} \in \mathbb{R}^{N+1}, \tag{6}$$

where $\odot$ represents element-wise multiplication and $\bar{\mathbf{Q}}^{t-1} = \prod_{\tau=1}^{t-1} \mathbf{Q}^\tau$ is the cumulative transition matrix up to step $t - 1$. The numerator is an unnormalized probability vector in $\mathbb{R}^{N+1}$, and the denominator is a

scalar normalization constant ensuring $\sum_{i=1}^{N+1} \mathbf{q}_v^{(t)}[i] = 1$. Although the node embeddings remain fixed, the selection probabilities $\mathbf{q}_v^{(t)}$ evolve over time through the transition dynamics.

*(3) Node selection:* During inference, starting from $\mathcal{V}_1 = \mathcal{V}$, at each step $t$, the node is selected by comparing the absorption probabilities to the masked state:

$$v_t^* = \arg\max_{v \in \mathcal{V}_t} \mathbf{q}_v^{(t)}[m], \tag{7}$$

where $\mathbf{q}_v^{(t)}[m]$ denotes the probability of node $v$ transitioning to the masked state $m = N + 1$ at step $t$. The active node set is then updated: $\mathcal{V}_{t+1} = \mathcal{V}_t \setminus \{v_t^*\}$. This process continues until all nodes are absorbed at step $N$, producing the final node order $\sigma = (v_1^*, v_2^*, \ldots, v_N^*)$. Since the node embeddings remain fixed throughout the ordering process, we denote them simply as $\{\mathbf{h}_v\}_{v \in \mathcal{V}}$ in downstream modules (dropping the superscript (0)). These embeddings, together with the learned order $\sigma$, serve as input to the subsequent tree construction and branch length estimation modules.

**Ordering Supervision Module** We supervise the Dynamic Ordering Network (DON) by aligning its predicted node absorption order with a reference order $\sigma^{\text{ref}} = (v_1^{\text{ref}}, v_2^{\text{ref}}, \ldots, v_N^{\text{ref}})$ obtained from external phylogenetic inference tools (e.g., MrBayes). During training, the active node set at each step $t$ is determined by the ground-truth prefix as described above (i.e., $\mathcal{V}_t = \mathcal{V} \setminus \{v_1^{\text{ref}}, \ldots, v_{t-1}^{\text{ref}}\}$), ensuring that DON learns to predict the next node conditioned on the correct absorption history. The supervision objective is the negative log-likelihood (NLL) of the ground-truth sequence:

$$\mathcal{L}_{\text{DON}} = -\sum_{t=1}^{N} \log p_t(v_t^{\text{ref}}), \quad \text{where} \quad p_t(v) = \frac{\exp(\mathbf{q}_v^{(t)}[m])}{\sum_{u \in \mathcal{V}_t} \exp(\mathbf{q}_u^{(t)}[m])}. \tag{8}$$

This objective encourages DON to assign high probability to biologically consistent construction sequences while maintaining differentiability for end-to-end training.

### 4.3 AutoRegressive Tree Construction with Dynamic Node Insertion

Given the biologically informed node ordering $\sigma = (v_1^*, v_2^*, \ldots, v_N^*)$ and contextual node embeddings $\{\mathbf{h}_v\}_{v \in \mathcal{V}}$ from the DON module, the autoregressive tree construction stage determines *where* each node should be inserted. For notational simplicity, we denote $v_i = v_i^*$ throughout this subsection. The embeddings are directly passed from DON without re-encoding, preserving their biological semantics, while the ordering $\sigma$ is used in two complementary ways: (i) as an explicit bias in insertion scoring to prioritize evolutionarily important nodes, and (ii) as a scheduling signal for dynamic parallel insertion.

**Selection & Placement Module** Instead of the standard sequential insertion (one node per step), we propose *dynamic parallel insertion* that processes multiple nodes simultaneously while adapting the batch size over time. We initialize the tree with the first three nodes: $\mathcal{V}_0^{\text{placed}} = \{v_1, v_2, v_3\}$, $\mathcal{E}^{(0)} = \{(v_1, p_0), (v_2, p_0), (v_3, p_0)\}$ where $p_0$ is an internal node with embedding $\mathbf{r}_{p_0} = \frac{1}{3}(\mathbf{h}_{v_1} + \mathbf{h}_{v_2} + \mathbf{h}_{v_3})$. At each iteration $t \in \{1, \ldots, T\}$ (distinct from message-passing steps in DON), we maintain the placed input nodes $\mathcal{V}_t^{\text{placed}}$, remaining nodes $\mathcal{R}_t = \mathcal{V} \setminus \mathcal{V}_t^{\text{placed}}$, current edges $\mathcal{E}^{(t)}$, and embeddings for all nodes in the current tree (initially $\mathbf{r}_v = \mathbf{h}_v$ for input nodes). We select a subset $\mathcal{U}_t \subseteq \mathcal{R}_t$ according to a cosine schedule:

$$\mathcal{U}_t = \text{SelectNodes}(\mathcal{R}_t, \rho_t), \quad \rho_t = \frac{1}{2}\left(1 + \cos\left(\frac{t-1}{T-1}\pi\right)\right), \tag{9}$$

where $\rho_t \in (0, 1]$ controls the fraction of remaining nodes to insert at iteration $t$. This schedule starts aggressive (inserting many nodes when the tree is sparse) and becomes conservative (inserting fewer nodes when the tree is complex), automatically balancing accuracy and efficiency. We terminate when $\mathcal{R}_t = \emptyset$, which typically occurs in $O(\sqrt{N})$ iterations. For each node $v \in \mathcal{U}_t$, we compute its contextual embedding by attending to nodes already in the tree: $\mathbf{r}_v = \text{MHA}(\mathbf{h}_v, \{\mathbf{r}_u : u \in \text{current tree}\})$. For each edge $e = (u_1, u_2) \in \mathcal{E}^{(t)}$, we compute edge embedding $\mathbf{r}_e = \frac{1}{2}(\mathbf{r}_{u_1} + \mathbf{r}_{u_2})$ and score:

$$s_{v,e} = \text{MLP}(\mathbf{r}_v, \mathbf{r}_e, \mathbf{r}_v \odot \mathbf{r}_e, \text{PE}(t)). \tag{10}$$

To explicitly leverage the biologically informed ordering, we introduce a rank-based bias. Let $\text{Rank}_\sigma(v) \in \{1, \ldots, N\}$ denote the position of $v$ in $\sigma$ (i.e., $\text{Rank}_\sigma(v_i) = i$).

$$\tilde{\mathbf{s}}_v = \mathbf{s}_v + \alpha \cdot (N - \text{Rank}_\sigma(v)) \cdot \mathbf{1}, \tag{11}$$

where $\mathbf{1} \in \mathbb{R}^{|\mathcal{E}^{(t)}|}$ is a vector of ones and $\alpha \in \mathbb{R}_+$ controls the strength of evolutionary guidance. This ensures that nodes prioritized by DON are favored during insertion, creating a direct pathway from learned evolutionary patterns to tree topology. The insertion edge is sampled from $e_v \sim \text{Multinomial}(\text{softmax}(\tilde{\mathbf{s}}_v))$.

**Node Update Module**   After sampling insertion edges $\{e_v\}_{v \in \mathcal{U}_t}$, we update the tree sequentially (following the order in $\sigma$) to avoid conflicts. For each $v \in \mathcal{U}_t$, if $e_v = (u_1, u_2)$ is in the current edge set, we remove $e_v$, create a new internal node $p[v]$, and add edges $(v, p[v])$, $(p[v], u_1)$, $(p[v], u_2)$. The new internal node's embedding is initialized as $\mathbf{r}_{p[v]} = \frac{1}{3}(\mathbf{r}_v + \mathbf{r}_{u_1} + \mathbf{r}_{u_2})$. If $e_v$ was already removed by a previous insertion in this batch, we resample $e_v$ from the updated edge set. We update $\mathcal{V}_{t+1}^{\text{placed}} = \mathcal{V}_t^{\text{placed}} \cup \mathcal{U}_t$ and repeat until all input nodes are placed. The final topology $\tau = (\mathcal{V}_\tau, \mathcal{E}_\tau)$ has $|\mathcal{V}_\tau| = 2N - 2$ nodes (including $N$ input nodes and $N - 2$ internal nodes) and $|\mathcal{E}_\tau| = 2N - 3$ edges. The node embeddings $\{\mathbf{r}_u\}_{u \in \mathcal{V}_\tau}$, together with the contextual embeddings $\{\mathbf{h}_v\}_{v \in \mathcal{V}}$ from DON, are passed to the branch length refinement module (Sec. 4.4) for metric estimation.

### 4.4 Dual-Pass Traversal for Branch Length Learning

The branch length learning module (Fig. 3) takes as input the inferred topology $\tau = (\mathcal{V}_\tau, \mathcal{E}_\tau)$, node embeddings $\{\mathbf{r}_u\}_{u \in \mathcal{V}_\tau}$ from the tree construction module, and the multiple sequence alignment $\mathcal{A}$. It jointly captures global and local structural cues via iterative dual-pass traversal with progressive branch length refinement, followed by graph-based encoding and differentiable sampling.

**Iterative Dual-Pass Aggregation**   We perform $K$ iterations (typically $K = 3$) of dual-pass aggregation to progressively refine branch length estimates. To enable tree traversal, we arbitrarily root the unrooted topology $\tau$ at the midpoint of an edge; since the final topology remains unrooted, the choice of root does not affect the branch length estimates. At each iteration $k \in \{1, \ldots, K\}$, we execute the following steps:

**(1) Postorder aggregation** propagates information bottom-up from leaves to root:

$$\mathbf{h}_u^{\text{fwd},(k)} = \begin{cases} \mathbf{r}_u, & \text{if } \mathcal{C}(u) = \emptyset \\ \text{GRU}\left(\mathbf{r}_u, \frac{1}{|\mathcal{C}(u)|} \sum_{v \in \mathcal{C}(u)} \phi(\mathbf{h}_v^{\text{fwd},(k)}, \ell_e^{(k)})\right), & \text{otherwise} \end{cases} \tag{12}$$

where $\mathbf{r}_u$ is the node embedding from tree construction (for both input nodes and internal nodes), $\mathcal{C}(u)$ is the child set of $u$, $\ell_e^{(k)} \in \mathbb{R}_+$ is the current branch length estimate (with $\ell_e^{(1)} = 0.1$), and $\phi : \mathbb{R}^d \times \mathbb{R} \to \mathbb{R}^d$ is an MLP.

**(2) Preorder aggregation** propagates refined context top-down from root to leaves:

$$\mathbf{h}_v^{\text{bwd},(k)} = \psi\left(\mathbf{h}_u^{\text{bwd},(k)}, \mathbf{h}_v^{\text{fwd},(k)}, \text{PE}(\text{depth}(v))\right), \tag{13}$$

where $\psi : \mathbb{R}^d \times \mathbb{R}^d \times \mathbb{R}^{d_{\text{PE}}} \to \mathbb{R}^d$ is an MLP, $\text{PE}(\cdot) \in \mathbb{R}^{d_{\text{PE}}}$ encodes the depth of node $v$, and $\mathbf{h}_{\text{root}}^{\text{bwd},(k)} = \mathbf{h}_{\text{root}}^{\text{fwd},(k)}$.

**(3) Branch length refinement.**   We combine bidirectional features $\mathbf{x}_i^{(k)} = \text{Concat}(\mathbf{h}_i^{\text{fwd},(k)}, \mathbf{h}_i^{\text{bwd},(k)})$. For intermediate iterations $(k < K)$, we use Branch Length Head (BLH), a lightweight MLP, to predict $b_e^{(k)} = \exp(\text{BLH}_{\text{iter}}(\mathbf{x}_u^{(k)}, \mathbf{x}_v^{(k)}))$ and set $\ell_e^{(k+1)} = b_e^{(k)}$ for the next iteration.

**Graph Encoding and Final Prediction**   After $K$ iterations, we pass the final features $\{\mathbf{x}_i^{(K)}\}_{i \in \mathcal{V}_\tau}$ through $L = 2$ layers of graph attention networks (Veličković et al., 2017) on $(\mathcal{V}_\tau, \mathcal{E}_\tau)$, yielding refined representations $\{\mathbf{z}_i\}_{i \in \mathcal{V}_\tau}$. A separate branch length head (a dual-output MLP) then produces final log-Gaussian parameters. Specifically, the head outputs $(\mu_e, \tilde{\sigma}_e) = \text{BranchHead}(\mathbf{z}_u, \mathbf{z}_v)$, and we set $\sigma_e = \exp(\tilde{\sigma}_e)$ to ensure positivity. Differentiable sampling is enabled via reparameterization:

$$b_e = \exp(\mu_e + \sigma_e \cdot \epsilon_e), \quad \text{where } \epsilon_e \sim \mathcal{N}(0, 1) \text{ (training)}, \ \epsilon_e = 0 \text{ (inference)}. \tag{14}$$

Table 1: Research Questions (RQs) and their corresponding sub-questions.

| | |
|---|---|
| **RQ1: Performance** | How well does MDTree perform in generating tree topologies (TDE) and inferring branch lengths (VBPI)? |
| **RQ2: Time Efficiency** | How efficient is MDTree in reducing runtime? |
| **RQ3: Tree Quality** | How optimal is MDTree to generate a tree structure? (RQ3-1) How diverse are the tree topologies generated by MDTree? (RQ3-2) How consistent is the MDTree-generated tree compared to MrBayes? (RQ3-3) |
| **RQ4: Module Impact** | How does each MDTree's module affect its performance? (RQ4-2) How do key hyper-parameters affect MDTree? (RQ4-2) |
| **RQ5: Case Study** | What evolutionary relationships between species does MDTree learn? |

The collection $\boldsymbol{\ell}_\tau = (b_e : e \in \mathcal{E}_\tau)$ forms the final branch length vector.

**Branch Length Loss** The predicted branch lengths $\boldsymbol{\ell}_\tau$ are supervised by the negative log-likelihood under a continuous-time Markov chain (CTMC) substitution model (Yang, 1994):

$$\mathcal{L}_{\mathrm{len}} = -\sum_{c=1}^{|\mathcal{A}|} \log p_{\mathrm{CTMC}}(\mathcal{A}_c \mid \tau, \boldsymbol{\ell}_\tau, \Theta_{\mathrm{sub}}), \tag{15}$$

where $p_{\mathrm{CTMC}}$ is computed via Felsenstein's pruning algorithm, $\mathcal{A}_c$ denotes the $c$-th column of the alignment, and $\Theta_{\mathrm{sub}} = \{\boldsymbol{\pi}, \mathbf{Q}\}$ represents the substitution model parameters (nucleotide frequencies $\boldsymbol{\pi}$ and rate matrix $\mathbf{Q}$) estimated from $\mathcal{A}$ via maximum likelihood (Yang, 1994). This iterative design progressively refines branch lengths through dual-pass aggregation, while the final reparameterized sampling enables stable gradient-based optimization.

### 4.5 MDTree Loss and Training

Building on the biologically informed node ordering (Sec. 4.2), topology construction (Sec. 4.3), and branch length estimation (Sec. 4.4), MDTree integrates these components via Variational Bayesian Phylogenetic Inference (VBPI).

**VBPI Objective** The joint posterior $q_{\theta,\phi}(\tau, \boldsymbol{\ell}_\tau) = q_\theta(\tau) q_\phi(\boldsymbol{\ell}_\tau \mid \tau)$ is optimized via the ELBO:

$$\mathcal{L}_{\mathrm{VBPI}} = \mathbb{E}_{q_\theta(\tau) q_\phi(\boldsymbol{\ell}_\tau|\tau)} \left[\log p(\mathcal{A} \mid \tau, \boldsymbol{\ell}_\tau)\right] - \mathrm{KL}(q_\theta(\tau) \,\|\, p(\tau)) - \mathbb{E}_{q_\theta(\tau)} \left[\mathrm{KL}(q_\phi(\boldsymbol{\ell}_\tau \mid \tau) \,\|\, p(\boldsymbol{\ell}_\tau \mid \tau))\right], \tag{16}$$

where $p(\mathcal{A} \mid \tau, \boldsymbol{\ell}_\tau)$ is the CTMC likelihood, and $p(\tau)$, $p(\boldsymbol{\ell}_\tau \mid \tau)$ are uniform and exponential priors.

**Unified Objective** The complete training objective combines node ordering supervision, VBPI, and auxiliary branch length loss:

$$\mathcal{L}_{\mathrm{MDTree}} = \lambda_{\mathrm{DON}} \mathcal{L}_{\mathrm{DON}} + \mathcal{L}_{\mathrm{VBPI}} + \lambda_{\mathrm{len}} \mathcal{L}_{\mathrm{len}}, \tag{17}$$

with $\lambda_{\mathrm{DON}} = 0.1$ and $\lambda_{\mathrm{len}} = 0.5$ (Eq. 8). This end-to-end coupling yields phylogenies that are both topologically accurate and metrically consistent.

## 5 Experiments

In this section, we demonstrate the effectiveness of our proposed MDTree in terms of the research questions in Table 1.

Table 2: **Comparison of KL divergence (↓) across eight benchmark datasets with different methods. Boldface** for the highest result, Underline for the second highest result of traditional methods. Results are reported as mean (standard deviation).

| Methods | Dataset (#Taxa,#Sites) | DS1 (27,1949) | DS2 (29,2520) | DS3 (36,1812) | DS4 (41,1137) | DS5 (50,378) | DS6 (50,1133) | DS7 (59,1824) | DS8 (64,1008) |
|---|---|---|---|---|---|---|---|---|---|
| | Sampled Trees | 1228 | 7 | 43 | 828 | 33752 | 35407 | 1125 | 3067 |
| | GT Trees | 2784 | 42 | 351 | 11505 | 1516877 | 809765 | 11525 | 82162 |
| MCMC-based | SBN | 0.0707 (0.0089) | 0.0144 (0.0021) | 0.0554 (0.0067) | 0.0739 (0.0095) | 1.2472 (0.1621) | 0.3795 (0.0493) | 0.1531 (0.0199) | 0.3173 (0.0412) |
| | SRF | 0.0155 (0.0023) | 0.0122 (0.0018) | 0.3539 (0.0460) | 0.5322 (0.0692) | 11.5746 (1.5047) | 10.0159 (1.3021) | 1.2765 (0.1659) | 2.1653 (0.2815) |
| | CCD | 0.6027 (0.0783) | 0.0218 (0.0032) | 0.2074 (0.0270) | 0.1952 (0.0254) | 1.3272 (0.1725) | 0.4526 (0.0588) | 0.3292 (0.0428) | 0.4149 (0.0539) |
| | SBN-SA | 0.0687 (0.0089) | 0.0218 (0.0032) | 0.2074 (0.0270) | 0.1952 (0.0254) | 1.3272 (0.1725) | 0.4526 (0.0588) | 0.3292 (0.0428) | 0.4149 (0.0539) |
| | SBN-EM | 0.0136 (0.0020) | 0.0199 (0.0029) | 0.1243 (0.0162) | 0.0763 (0.0099) | 0.8599 (0.1118) | 0.3016 (0.0392) | 0.0483 (0.0063) | 0.1415 (0.0184) |
| | SBN-EM-$\alpha$ | 0.0130 (0.0019) | 0.0128 (0.0019) | 0.0882 (0.0115) | 0.0637 (0.0083) | 0.8218 (0.1068) | 0.2786 (0.0362) | 0.0399 (0.0052) | 0.1236 (0.0161) |
| Structure Generation | ARTree | 0.0045 (0.0007) | **0.0097** (0.0012) | 0.0548 (0.0071) | 0.0299 (0.0039) | 0.6266 (0.0815) | 0.2360 (0.0307) | 0.0191 (0.0025) | 0.0741 (0.0096) |
| | **MDTree** | **0.0036** (0.0005) | 0.0129 (0.0016) | **0.0446** (0.0055) | **0.0216** (0.0026) | **0.5751** (0.0690) | **0.1591** (0.0191) | **0.0169** (0.0020) | **0.0634** (0.0076) |

## 5.1 Experiment Setup

**Evaluation Tasks and Datasets.** We assess MDTree's performance on two key tasks: TDE, which focuses on optimizing tree topologies with MLL metric, and VBPI, where tree topologies and branch lengths are jointly inferred, using ELBO and MLL. These evaluations span eight diverse benchmark datasets, covering various organisms like marine animals, plants, bacteria, fungi, and eukaryotes, as outlined in Appendix C.

**Baselines.** MDTree is compared against three primary groups of baselines: (1) MCMC-based methods (e.g., MrBayes, SBN), (2) Structure Representation methods (VBPI, VBPI-GNN), which leverage pre-generated topologies, and (3) Structure Generation methods for Bayesian inference without pre-selected topologies. Notably, ARTree, a comparable autoregressive method like ours, is highlighted for comparison. All training details and hyperparameters are provided in Appendix E.

## 5.2 Comparison Results on Benchmarks (RQ1)

**The TDE Task.** We compare the KL divergence to measure the difference between the model's generated tree topology distribution $q_\theta(\tau)$ and the true posterior $p(\tau)$: $\mathrm{KL}(p(\tau)||q_\theta(\tau)) = \sum_\tau p(\tau) \log \frac{p(\tau)}{q_\theta(\tau)}$. Table 2 shows that our MDTree consistently achieves lower KL divergence across all datasets compared to MCMC-based and structure generation methods. On complex datasets such as DS5 and DS6, it outperforms ARTree and SBN, demonstrating superior scalability. Even on smaller datasets like DS1 and DS3, the performance remains competitive, highlighting the model's robustness. The comparison with ARTree underscores the advantage of autoregressive models, including ours, particularly on larger, more complex datasets.

**The VBPI Task.** We evaluate the VBPI task using ELBO and MLL metrics. Since direct computation of MLL is intractable, it is approximated via importance sampling. Unlike TDE, which relies on known tree topologies, VBPI evaluates the fit between model-generated tree topologies and branch lengths and the observed gene sequence data. As shown in Table 3 and Table 5, Tree Structure Generation methods exhibit broader applicability in MLL and ELBO metrics compared to Structure Representation methods, which are restricted by their reliance on pre-generated topologies. Our method, MDTree, consistently achieves the highest metrics across all datasets, highlighting its enhanced capacity to approximate the posterior distribution of tree topologies and branch lengths. Fig. 4 shows MDTree's superior stability and fast convergence in ELBO on DS1, outperforming baselines. ARTree and SBN improve later but with fluctuations, while Geo-

Table 3: **Evaluation of MLL (↑) on eight benchmark datasets.** VBPI and VBPI-GNN utilize pre-generated tree topologies during training, making **direct comparisons challenging**. **Boldface** highlights the highest result, Text denotes the second highest of structure generation methods, and Text indicates the second highest of MCMC-based methods. Results are reported as mean (standard deviation).

| Methods | Dataset (#Taxa,#Sites) | DS1 (27,1949) | DS2 (29,2520) | DS3 (36,1812 ) | DS4 (41,1137) | DS5 (50,378) | DS6 (50,1133) | DS7 (59,1824) | DS8 (64,1008) |
|---|---|---|---|---|---|---|---|---|---|
| MCMC-based | MrBayes | -7108.42 (0.18) | -26367.57 (0.48) | -33735.44 (0.50) | -13330.44 (0.54) | -8214.51 (0.28) | -6724.07 (0.86) | -37332.76 (2.42) | -8649.88 (1.75) |
| | SBN | -7108.41 (0.15) | -26367.71 (0.08) | -33735.09 (0.09) | -13329.94 (0.20) | -8214.62 (0.40) | -6724.37 (0.43) | -37331.97 (0.28) | -8650.64 (0.50) |
| Structure Representation | VBPI | -7108.42 (0.10) | -26367.72 (0.12) | -33735.10 (0.11) | -13329.94 (0.31) | -8214.61 (0.67) | -6724.34 (0.68) | -37332.03 (0.43) | -8650.63 (0.55) |
| | VBPI-GNN | -7108.41 (0.14) | -26367.73 (0.07) | -33735.12 (0.09) | -13329.94 (0.19) | -8214.64 (0.38) | -6724.37 (0.40) | -37332.04 (0.12) | -8650.65 (0.45) |
| Structure Generation | ARTree | -7108.41 (0.19) | -26367.71 (0.07) | -33735.09 (0.09) | -13329.94 (0.17) | -8214.59 (0.34) | -6724.37 (0.46) | -37331.95 (0.27) | -8650.61 (0.48) |
| | phi-CSMC | -7290.36 (7.23) | -30568.49 (31.34) | -33798.06 (6.62) | -13582.24 (35.08) | -8367.51 (8.87) | -7013.83 (16.99) | NA | -9209.18 (18.03) |
| | GeoPhy | -7111.55 (0.07) | -26379.48 (11.60) | -33757.79 (8.07) | -13342.71 (1.61) | -8240.87 (9.80) | -6735.14 (2.64) | -37377.86 (29.48) | -8663.51 (6.85) |
| | GeoPhy LOO(3) | -7116.09 (10.67) | -26368.54 (0.12) | -33735.85 (0.12) | -13337.42 (1.32) | -8233.89 (6.63) | -6735.9 (1.13) | -37358.96 (13.06) | -8660.48 (0.78) |
| | PhyloGFN | -7108.95 (0.06) | -26368.90 (0.28) | -33735.60 (0.35) | -13331.83 (0.19) | -8215.15 (0.20) | -6730.68 (0.54) | -37359.96 (1.14) | -8654.76 (0.19) |
| | **Ours** | **-7101.38** (0.07) | **-26357.96** (0.06) | **-33715.31** (0.10) | **-13322.10** (1.34) | **-8210.76** (0.23) | **-6713.13** (0.32) | **-37326.50** (1.39) | **-8645.07** (0.69) |

Table 4: **Comparison of mean log-likelihood (MLL) and runtime between ARTree and MDTree under RWS and VIMCO optimization, each trained for 400,000 iterations.** MDTree consistently achieves higher MLL and reduces runtime by over 40% compared to ARTree.

| Methods | MLL | Runtime (s) |
|---|---|---|
| ARTree_rws | -7107.74 | 128.7 |
| **MDTree_rws** | -7103.71 | 75.0(↓41.72%) |
| ARTree_vimco | -7106.59 | 114.7 |
| **MDTree_vimco** | -7101.38 | 63.7(↓44.46%) |

Phy performs the worst with consistently low and unstable values. Fig. 5 highlights MDTree's advantages in MLL, quickly reaching and maintaining high scores, whereas ARTree, SBN, and especially GeoPhy lag behind.

Table 5: **Evaluation of ELBO (↑) on eight datasets.** Higher values indicate better performance. Results for GeoPhy were not reported in its original publication and are reproduced by us. Light gray marks the best baseline result, and darker gray marks the best overall result. Our method consistently achieves the highest ELBO across all datasets.

| Methods | Dataset (#Taxa,#Sites) | DS1 (27,1949) | DS2 (29,2520) | DS3 (36,1812 ) | DS4 (41,1137) | DS5 (50,378) | DS6 (50,1133) | DS7 (59,1824) | DS8 (64,1008) |
|---|---|---|---|---|---|---|---|---|---|
| MCMC-based | SBN | -7110.24 (0.03) | -26368.88 (0.03) | -33736.22 (0.02) | -13331.83 (0.02) | -8217.80 (0.04) | -6728.65 (0.04) | -37334.85 (0.03) | -8655.05 (0.04) |
| Structure Generation | ARTree | -7110.09 (0.04) | -26368.78 (0.07) | -33735.25 (0.08) | -13330.27 (0.05) | -8215.34 (0.04) | -6725.33 (0.06) | -37332.54 (0.13) | -8651.73 (0.05) |
| | GeoPhy | -7116.67 (1.71) | -26434.84 (0.10) | -33766.72 (0.15) | -13389.36 (3.45) | -8220.91 (2.64) | -6769.41 (3.25) | -37882.96 (1.97) | -8654.39 (0.97) |
| | Ours | **-7005.98** (0.06) | **-26362.75** (0.12) | **-33430.94** (0.34) | **-13113.03** (3.65) | **-8053.23** (2.57) | **-6324.90** (1.26) | **-36838.42** (1.99) | **-8409.06** (1.09) |

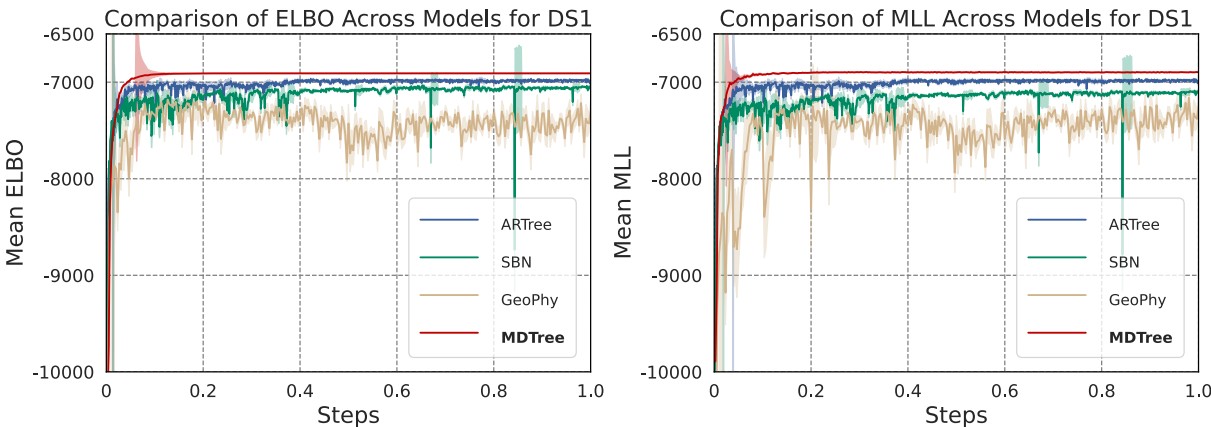

Figure 4: Comparison of ELBO.    Figure 5: Comparison of MLL.

### 5.3  Runtime Reduction and Efficiency Evaluation (RQ2)

MDTree demonstrates substantial runtime efficiency across all datasets, outperforming ARTree consistently. Both runtime and the number of nodes are log-transformed on the vertical axes, with solid and dashed lines representing the RWS and VIMCO optimization techniques. MDTree achieves faster than ARTree across all datasets, with VIMCO providing further reductions, especially for MDTree-VIMCO, which exhibits the lowest runtime. The efficiency of MDTree becomes even more apparent as dataset complexity increases. Table 4 confirms this finding, with MDTree reducing runtime by 41.72% (RWS) and 44.46% (VIMCO) compared to ARTree while maintaining superior MLL metrics. This underscores MDTree's efficiency and scalability, particularly with VIMCO optimization.

### 5.4  Tree Parsimony in Phylogenetic Inference (RQ3-1)

To evaluate the parsimony of tree structures generated by the model, we follow established methodologies (Zhou et al., 2024), minimizing the genetic mutations required to infer the optimal tree. The parsimony score evaluates how well the generated tree adheres to the principle of minimizing evolutionary changes, where fewer mutations are assumed to explain the observed genetic data better. We compare the results against the most parsimonious tree identified by the traditional PAUP* tool (Swofford, 1998). The parsimony score in Fig. 6 denotes the minimum mutations of genetic changes needed to account for the evolutionary relationships in the data. Since scores are plotted as negative values, lower scores indicate more complex trees and, consequently, poorer model performance. MDTree and ARTree achieved higher scores (approaching -4000) in fewer steps, reflecting simpler and more parsimonious trees. In contrast, PhyloGFN exhibited early fluctuations and ultimately stabilized around -5000, indicating suboptimal performance compared to others.

### 5.5  Tree Topological Diversity in Generated Trees (RQ3-2)

To assess the diversity of tree topologies generated by MDTree, we use three metrics: Simpson's Diversity Index (He & Hu, 2005), Top Frequency, and Top 95% Frequency, as detailed in Table 6. A higher Diversity Index, which approaches 1, suggests broad diversity among generated tree topologies. A larger number of topologies in the Top 95% Frequency implies the generated trees are more varied and distributed across many unique structures. Conversely, a lower Top Frequency suggests the absence of a dominant tree structure, pointing toward a more balanced generation. For instance, in DS3, with 36 species sequences, the Top 95% Topologies metric reveals 1,146 distinct tree structures, indicating a wide range of possible phylogenetic solutions. MDTree achieves a Diversity Index close to 1, showcasing its capacity for generating highly diverse topologies even in complex datasets. Furthermore, the Top Frequency metric remains notably low, further reinforcing the diversity and indicating that no single tree topology is overly dominant.

Table 6: **Topological comparison of three tree diversity metrics. Higher** values of Simpson's Diversity Index and the number of topologies accounting for the top 95% cumulative frequency indicate better diversity. In contrast, a **lower frequency** of the most frequent topology reflects a balanced distribution.

| Dataset | Statistics | MrBayes | ARTree | Ours |
|---------|-----------|---------|--------|------|
| DS1 | Diversity Index (↑) | 0.87 | 0.89 | **0.99** |
|     | Top Frequency (↓) | 0.27 | 0.1 | **0.007** |
|     | Top 95% Frequency (↑) | 42 | 10 | **121** |
| DS2 | Diversity Index (↑) | 0.89 | 0.96 | **0.99** |
|     | Top Frequency (↓) | 0.27 | 0.43 | **0.13** |
|     | Top 95% Frequency (↑) | 208 | 203 | **301** |
| DS3 | Diversity Index (↑) | 0.98 | 0.89 | **0.90** |
|     | Top Frequency (↓) | 0.02 | 0.01 | **0.004** |
|     | Top 95% Frequency (↑) | 753 | 509 | **1146** |
| DS4 | Diversity Index (↑) | 0.86 | 0.89 | **0.99** |
|     | Top Frequency (↓) | 0.11 | 0.05 | **0.002** |
|     | Top 95% Frequency (↑) | 4169 | 4125 | **8746** |

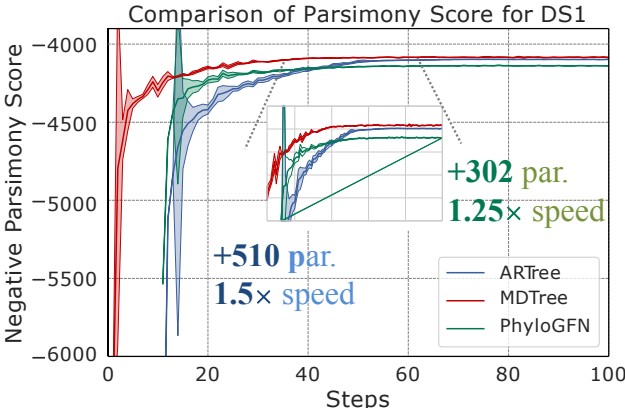

Figure 6: **Comparison of negative parsimony scores on the DS1 dataset.** The parsimony score denotes the minimum number of variation steps required to interpret each tree. The lower the negative score, the poorer the model performance.

## 5.6 Bipartition Frequency for Tree Quality (RQ3-3)

In phylogenetic analysis, a bipartition refers to dividing taxa (species or genes) into two groups on either side of a node within the tree. When multiple tree samples are generated, as in Bayesian inference methods like MrBayes, each sample may have a different topology. Bipartition frequency quantifies how often a specific bipartition appears across all tree samples, providing insight into the support for particular evolutionary relationships. We use this bipartition frequency distribution to assess the model's ability to capture phylogenetic relationships, as shown in Fig. 7. The horizontal axis indicates the bipartition rank within the tree topology, while the vertical axis displays the normalized occurrence frequency of each bipartition. The MDTree and MrBayes **curves are closely aligned**, indicating that MDTree's results closely match those of the widely accepted gold standard. In contrast, the ARTree method shows a noticeable deviation, especially in the higher-ranked bipartitions, demonstrating that MDTree offers improved accuracy over ARTree in capturing evolutionary structures. This suggests that MDTree captures the evolutionary patterns with greater accuracy compared to ARTree.

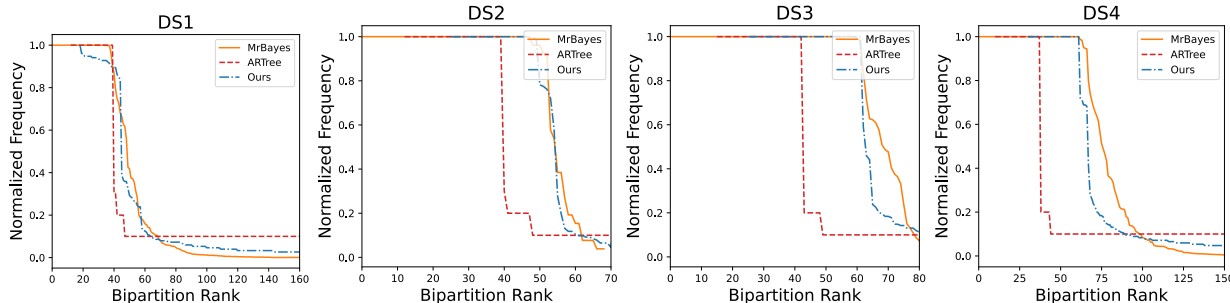

Figure 7: Bipartition frequency distribution of tree topologies. **The closer the two curves are, the better.**

Table 7: Comparison of different genomic language models (LMs) as structure generators in our framework, evaluated on Mean Log-Likelihood (MLL, ↑) and Evidence Lower Bound (ELBO, ↑). Models include DNABERT2, HyenaDNA, and NT. Higher values indicate better performance. DNABERT2 achieves the highest MLL and ELBO among the tested models, indicating its superior ability to capture genomic sequence patterns beneficial for phylogenetic inference.

| Method | MLL(↑) | ELBO(↑) |
|---|---|---|
| DNABERT2 | **-7101.38** | **-7005.98** |
| HyenaDNA | -7109.36 | -7014.17 |
| NT | -7111.07 | -7017.11 |

### 5.7 Analysis and Ablation (RQ4-1)

We compare MDTree with three other schemes, yielding the following observations: (i) Removing optimization techniques like RWS or VIMCO led to a performance drop of 5.21 in MLL, as shown by slight fluctuations in the MLL curve in Fig. 9, highlighting their role in stabilizing convergence. (ii) Excluding the LAX model of VIMCO optimization caused a decrease of 2.36 in MLL and 1.88 in ELBO, indicating its effectiveness in reducing variance during discrete sampling. (iii) Table 7 and Table 8 show that the removal of the DON results in the most significant impact, with a drop of about 3.67 in MLL, underscoring its critical role in optimizing node addition order and improving tree generation. Overall, the full MDTree consistently achieves the best across both metrics. We select the genome-specific foundation model DNABERT2 for our phylogenetic inference research. Although models like HyenaDNA (Nguyen et al., 2023) and Nucleotide Transformer (NT) (Dalla-Torre et al., 2023) excel in long-sequence modeling, they are less apt for our specific needs. As shown in Table 7, DNABERT2 outperforms others, likely due to its specific optimization for genomic data.

### 5.8 Visualization of PhyloTree Structure on Real-World Data (RQ5)

To assess the biological relevance of the tree structure generated by MDTree, we applied it to construct a phylogenetic tree for an Angiosperms353 genomic dataset (Zuntini et al., 2024). The tree successfully recovered major branches within the order Rosales, revealing distinct evolutionary lineages, including Rosaceae, Moraceae, and Polygonaceae families. As shown in Fig. 10, the genera Polygala vulgaris and Polygala balduinii are clearly separated from other groups, consistent with their classification in the Potentillaceae family. The remaining groups, distinguished by color, represent genera within the Rosaceae and Moraceae families, such as Rosa, Rubus, Ficus, and Adansonia. In Rosaceae, genera like Rosa, Rubus, and Prunus highlight their common evolutionary ancestry, while in Moraceae, Ficus and Broussonetia reflect the internal diversity and evolutionary divergence within the family.

Figure 8: **Ablation study of MDTree on four datasets, reported in mean log-likelihood (MLL) and ELBO (higher is better).** We evaluate the impact of removing the optimization phase, removing LAX in VIMCO, and removing the Dynamic Ordering Network (DON). The last column shows the average MLL across datasets, with green values indicating the drop compared to the full MDTree.

| Method | DS1 | | DS2 | | DS3 | | DS4 | | Average |
|---|---|---|---|---|---|---|---|---|---|
| | MLL | ELBO | MLL | ELBO | MLL | ELBO | MLL | ELBO | |
| MDTree | -7101.38 | -7005.98 | -26357.96 | -26362.75 | -33715.31 | -33430.94 | -13322.10 | -13113.03 | **-20051.18** |
| w/o optimization | -7106.59 | -7010.34 | -26371.02 | -26374.01 | -33733.25 | -33447.94 | -13339.71 | -13130.01 | -20064.11 (-12.93) |
| w/ vimco w/o Lax | -7103.74 | -7007.86 | -26361.81 | -26368.52 | -33718.20 | -33436.07 | -13326.95 | -13118.60 | -20055.22 (-4.04) |
| w/o DON | -7105.05 | -7010.02 | -26366.47 | -26372.04 | -33723.67 | -33439.18 | -13332.38 | -13121.33 | -20058.77 (-7.59) |

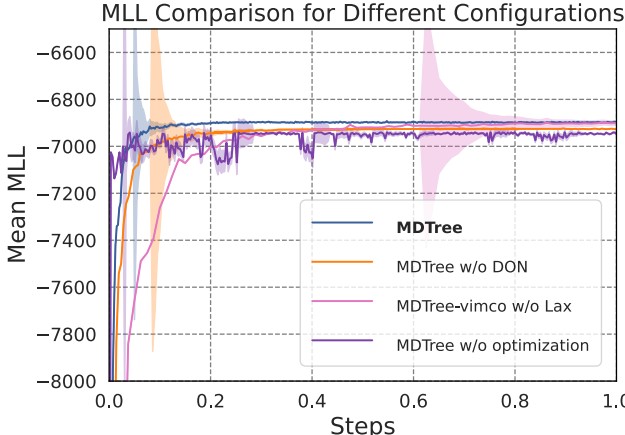

Figure 9: Ablation of different modules. MDTree w/o optimization curve exhibits **slight fluctuations**, emphasizing the importance of **optimization techniques** in improving stability.

# 6 Conclusion and Limitation

In this paper, we present MDTree, a novel framework that redefines phylogenetic tree generation as a Dynamic Autoregressive Tree Generation task. By leveraging a Diffusion Ordering Network to learn biologically informed node orders directly from genomic sequences, MDTree overcomes the limitations of fixed or random node orders. It integrates GNNs and Language Models to capture complex tree topologies, while a Dynamic Masking Mechanism enables parallel node processing, improving computational efficiency. Experiments on phylogenetic benchmarks show MDTree achieves state-of-the-art performance.

MDTree has yet to be applied to other sequence types, such as protein sequences. Future work will explore multimodal approaches, integrating genomic and protein data for more comprehensive evolutionary tree construction, as well as scaling the model for complex evolutionary scenarios.

### Author Contributions

Zelin Zang and Chenrui Duan contributed equally to this research. Z.L., J.W., J.L., Z.Lei and S.Z.L. conceived and supervised the project. Z.Z. and C.D. designed the model and C.D. conducted code and experiments. Z.Z., C.D., J.W., J.L., Z.Lei and S.Z.L. wrote the manuscript with input from all authors.

### Acknowledgments

This work was supported in part by National Science and Technology Major Project (No. 2022ZD0115101), National Natural Science Foundation of China Project (No. U21A20427), Project (No. WU2022A009) from the Center of Synthetic Biology and Integrated Bioengineering of Westlake University and InnoHK Program. We thank the AI Station of Westlake University for the support of GPUs.

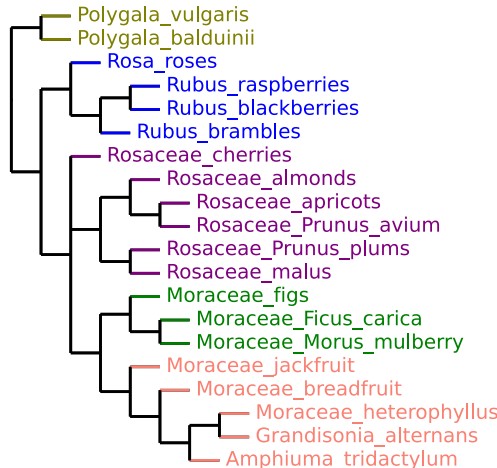

Figure 10: **Visualization of a generated phylogenetic tree for a subset of species from the Angiosperms353 dataset.** Different colors indicate distinct plant families or genera, illustrating the model's ability to cluster related species into coherent subtrees. For example, species within the genus *Rubus* (blue) and family *Moraceae* (green) are correctly grouped together, reflecting biologically plausible evolutionary relationships. This demonstrates that the proposed method can recover meaningful phylogenetic structure consistent with known taxonomy.

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
