## Supplement Material

## A    Background

### A.1    Phylogeny and Machine Learning Applications in Biology

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

## B  Related Work

Phylogenetic inference methods can be broadly categorized into two major classes: traditional methods and deep learning-based methods. Each class can be further divided into graph structure generation and representation models. In this section, we review these approaches in detail.

### B.1  Traditional Methods

Traditional phylogenetic inference methods primarily rely on predefined evolutionary models and statistical inference techniques. These methods typically assume specific evolutionary processes and use statistical approaches to search and optimize within a given tree structure space. They can be classified into graph structure generation and representation models.

**Graph Structure Generation Models:** MrBayes Ronquist et al. (2012) generates phylogenetic trees using Bayesian inference, estimating posterior probabilities based on sample relative frequency (SRF). However, the high-dimensional combinatorial space poses accuracy challenges, particularly for low-probability trees, requiring large sample sizes for stability. VaiPhy Koptagel et al. (2022) introduces the SLANTIS sampling strategy Diaconis (2019) to generate tree structures by learning phylogenetic tree topologies. This approach combines basic biological models, such as the JC model, to estimate branch lengths, producing more accurate tree structures.

**Graph Structure Representation Models:** SBN (Structured Bayesian Networks) Zhang & Matsen IV (2018a) focuses on learning the probability distribution of tree topologies from existing phylogenetic trees. By modeling subsplit relationships within a given set of trees, SBN captures the probabilistic structure of the entire tree space without directly estimating branch lengths. VBPI (Variational Bayesian Phylogenetic Inference) Zhang & Matsen IV (2018b) builds on the tree topology probability distributions provided by SBN, using variational inference to estimate the posterior distribution of tree structures. This method further optimizes branch lengths, offering a precise approximation of the posterior distribution.

While traditional methods provide a solid theoretical foundation, they often struggle with the complexity of high-dimensional data and intricate evolutionary relationships. The emergence of deep learning has introduced new approaches to address these challenges.

### B.2  Deep Learning-Based Methods

In recent years, deep learning techniques have demonstrated significant potential in phylogenetic inference, especially when dealing with complex, high-dimensional genomic data. These methods excel in generating and representing phylogenetic trees by learning latent representations or structural features from the data. They can be categorized into graph structure generation and representation models.

**Graph Structure Generation Models:**

- **Bayesian Generative Models (e.g., VAE):** These models learn latent representations of graphs using variational inference, from which new tree structures can be sampled. GeoPhy Mimori & Hamada (2024) exemplifies this approach by leveraging VAE to model the latent space of phylogenetic trees, generating diverse structures that accommodate complex evolutionary histories.

- **Autoregressive Models:** Autoregressive models generate tree structures incrementally, making them suitable for tasks with well-defined sequences or hierarchies. ARTree Xie & Zhang (2024)

employs a graph autoregressive model to generate detailed topologies, with branch lengths independently estimated using classical evolutionary models.

- **Diffusion Models:** Although diffusion models have not been widely applied in phylogenetic tree generation, our study integrates diffusion models with autoregressive models to generate the node addition order, enhancing the accuracy of tree structures. This demonstrates the potential of diffusion models in high-quality phylogenetic inference.

- **Generative Flow Networks (GFlowNets):** As illustrated by PhyloGFN Zhou et al. (2024), GFlowNets Hu et al. (2023) combined with Markov decision processes optimize the generation path, progressively constructing complex phylogenetic tree structures.

**Graph Structure Representation Models:** VBPI-GNN Zhang (2023) leverages pre-generated candidate tree structures and SBN-provided tree topology probability distributions, combined with variational inference, to optimize branch lengths and tree topologies, ultimately providing a precise approximation of the posterior distribution.

## C  Datasets

Our model, MDTree, conducts phylogenetic inference on biological sequence datasets comprising 27 to 64 species, as compiled in Lakner et al. (2008). Importantly, our approach does not require sequences to be of uniform length, thereby addressing a common limitation in traditional phylogenetic analyses. Table A1 summarizes the statistics of the benchmark datasets.

Table A1: Statistics of the benchmark datasets from DS1 to DS8..

| Dataset | # Species | # Sites | Reference |
|---------|-----------|---------|-----------|
| DS1 | 27 | 1949 | Hedges et al. (1990) |
| DS2 | 29 | 2520 | Garey et al. (1996) |
| DS3 | 36 | 1812 | Yang & Yoder (2003) |
| DS4 | 41 | 1137 | Henk et al. (2003) |
| DS5 | 50 | 378 | Lakner et al. (2008) |
| DS6 | 50 | 1133 | Zhang & Blackwell (2001) |
| DS7 | 59 | 1824 | Yoder & Yang (2004) |
| DS8 | 64 | 1008 | Rossman et al. (2001) |

## D  Method

**Calculation of the number of unlabelled nodes in DON.** The number of nodes unmasked at each step is dynamically determined by a mask rate modulated by a cosine function. Given a total of $T$ steps and $U$ nodes to be unmasked per step, the proportion of nodes to be unmasked at each step $t$ is computed as follows: $r_t = \frac{t}{T}, t = 1, 2, \ldots, T$. This is modulated by a cosine function to produce the mask rate: $\text{mask\_rate}_t = \cos\left(\frac{\pi}{2} \cdot r_t\right)$, where $\text{mask\_rate}_t$ controls the relative number of nodes unmasked at step $t$. The final number of nodes unmasked at each step is normalized to ensure that the total number of unmasked nodes across all steps sums to $T \times U$: $\text{unmasked\_nodes}_t = \left\lfloor \frac{\text{mask\_rate}_t}{\sum_{t=1}^{T} \text{mask\_rate}_t} \cdot T \cdot U \right\rfloor$, where $\lfloor \cdot \rfloor$ denotes rounding to the nearest integer.

# E  Experiment

## E.1  Training Details

We focus on the most challenging aspect of the phylogenetic tree inference task: the joint learning of tree topologies and branch lengths. For this, we employ a uniform prior for the tree topology and an independent and identically distributed (i.i.d.) exponential prior (Exp(10)) for the branch lengths. We evaluate all methods across eight real datasets (DS1-8) frequently used to benchmark phylogenetic tree inference methods. These datasets include sequences from 27 to 64 eukaryote species, each comprising 378 to 2520 sites. For our Monte Carlo simulations, we select $K = 2$ samples and apply an annealed unnormalized posterior during each $i$-th iteration, where $\lambda_n = \min\{1.0, 0.001 + i/H\}$ acts as the inverse temperature. This parameter starts at 0.001 and gradually increases to 1 over $H$ iterations, effectively simulating a cooling schedule commonly used in annealing algorithms, similar to the approach in Zhang & Matsen IV (2018a), with an initial temperature of 0.001, which gradually decreases over 100,000 steps.

During the model training process, we utilize stochastic gradient descent to process a total of one million Monte Carlo samples, employing $K$ samples at each training step. The stepping-stone (SS) algorithm Xie et al. (2011) in MrBayes is viewed as the gold-standard value. All models were implemented in Pytorch Paszke et al. (2019) with the Adam optimizer Kingma & Ba (2014). The MLL estimate is derived by sampling the importance of 1000 samples, with the larger mean value being better. The learning rate is initially set to 1e-4 and is reduced by 0.75 every 200,000 training steps. Momentum is set at 0.9 to prevent the optimization process from becoming trapped in local minima. Utilizing the StepLR scheduler, the current learning rate is multiplied by 0.75 every 200,000 steps to ensure steady progression, detailed in Tab. A2.

Table A2: Training Settings of MDTree.

| Training Configuration | |
| --- | --- |
| Optimizer | Adam optimizer |
| Learning rate | 1e-4 |
| Schedule | Step Learning Rate |
| Weight Decay | 0.0 |
| momentum | 0.9 |
| base_lr | 1e-4 |
| max_lr | 0.001 |
| scheduler.gamma | 0.75 |
| annealing init | 0.001 |
| annealing steps | 400,000 |

Table A3: Common Hyperparameters for MDTree.

| DON | |
| --- | --- |
| Hidden Dim. | 32 |
| # Layer | 2 |
| Output Dim. | 1 |
| TreeEncoder | |
| Hidden Dim. | 100 |
| # Heads | 4 |
| DGCNN | |
| # Layer | 2 |

## E.2  Hyper-Parameter Analysis (RQ4-2)

Table A4 summarizes the hyperparameter search results for the DON hidden dimension, Tree Network (Transformer) hidden dimension, and the number of attention heads. When increasing the number of heads from 1 to 4, ELBO improves from -7517.98 to -7005.98, and MLL improves from -7333.14 to -7101.38, demonstrating that more attention heads allow the model to capture richer dependencies. For the DON hidden dimension, a value of 32 achieves the best results, with an ELBO of -7005.98 and MLL of -7101.38. Similarly, tuning the Tree hidden dimension shows that 100 is optimal, yielding an ELBO of -7005.98 and MLL of -7101.38, while further increasing the dimension does not result in better performance. These results highlight the importance of tuning the number of heads and hidden dimensions to balance model complexity and generalization.

# F  Biological Interpretation of Generated Trees

Figure A1 illustrates the phylogenetic relationships among 20 fungal species from the DS10 dataset, generated using our proposed method. The tree structure reveals distinct clades corresponding to major taxonomic

Table A4: Hyperparameter Analysis of MDTree Performance.

| Configurations | Parameters | | | |
|---|---|---|---|---|
| | 1 | 2 | 3 | 4 |
| DON_hd=32, Tree_hd=100 | | | | |
| # Heads | 1 | 2 | 3 | 4 |
| ELBO | -7517.98 | -7111.95 | -7106.65 | **-7005.98** |
| MLL | -7333.14 | -7116.65 | -7104.82 | **-7101.38** |
| # Heads=4, Tree_hd=100 | | | | |
| DON_hidden dim | 8 | 16 | 32 | 64 |
| ELBO | -7016.75 | -7011.16 | **-7005.98** | -7013.96 |
| MLL | -7113.84 | -7117.83 | **-7101.38** | -7105.18 |
| # Heads=4, DON_hd=32 | | | | |
| Tree_hidden dim | 500 | 100 | 150 | 200 |
| ELBO | -7013.71 | **-7005.98** | -7012.07 | -7008.93 |
| MLL | -7112.71 | **-7101.38** | -7102.05 | -7121.51 |

groups, such as Ascomycota and Basidiomycota, highlighting evolutionary divergences. Notably, species within the same genus cluster closely together, reflecting their shared evolutionary history. The branch lengths indicate varying degrees of genetic divergence, with longer branches suggesting more significant evolutionary changes. This phylogenetic tree not only aligns with established biological classifications but also provides insights into the evolutionary trajectories of these fungal species, demonstrating the effectiveness of our method in capturing complex evolutionary relationships.

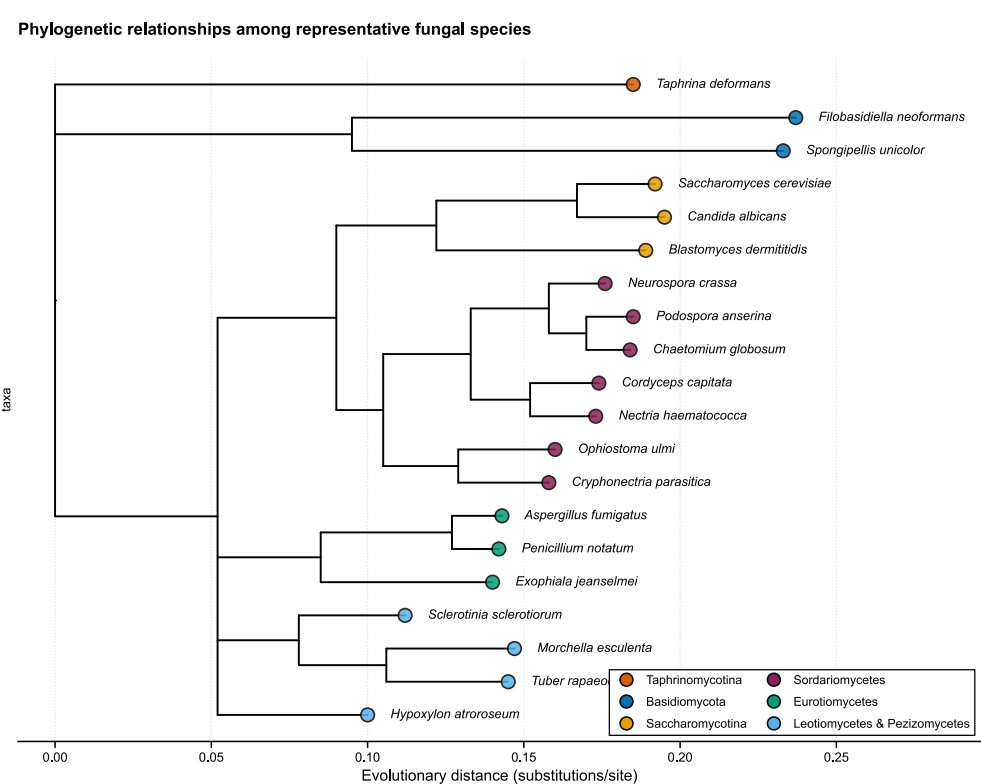

Figure A1: Phylogenetic relationships of 20 fungal species from DS10 dataset. Phylogenetic tree showing evolutionary distances among representative fungal species. Colors indicate major taxonomic groups. Tree generated using our proposed method.