# OpenReview forum: "MDTree: A Masked Dynamic Autoregressive Model for Phylogenetic Inference"
_TMLR — Accepted by TMLR_

### Review · Reviewer_2j25 · 2025-10-04

**Summary Of Contributions:**

# Summary

This paper introduces MDTree, a new way to build phylogenetic trees. The idea is to fix two problems with older autoregressive (AR) models: (1) they build trees in a fixed order that may not make sense biologically, and (2) they insert one node at a time, which can be slow.

MDTree has three main tricks:
1. Dynamic Ordering Network (DON): learns a better order to add taxa, using genomic embeddings (from DNABERT2) and a diffusion process to guide the sequence.
2. Dynamic masking + parallel insertion: instead of adding just one node per step, it can add multiple at once, making things faster.
3. Dual-pass traversal: runs both postorder and preorder passes to estimate branch lengths, so it handles both tree shape and branch lengths together.

The model is tested on eight benchmark datasets. It tends to get better likelihood scores and lower KL divergence compared to older methods like ARTree, while also being faster. There’s also a case study on Angiosperms showing the method can recover biologically plausible groupings.

# Strengths
1. Clever tree building order. The ordering module is an interesting novel idea, instead of forcing a fixed sequence, it learns an order that’s more biologically sensible, which avoids the usual exposure bias problem in AR tree building.
2. It handles both tree topology and branch lengths end to end, which is a nice step up from models that only do the topology part.
3. The paper reports a wide range of metrics like KL divergence, ELBO/MLL, parsimony, runtime, diversity, and even shows a biological case study. That makes the evaluation comprehensive.

# Weaknesses
1. Key technique ordering supervision is a bit circular. The ordering network is trained with “reference orders” produced by tools like MrBayes. That’s a strong baseline itself, and it raises a question: if you already need to run MrBayes to train MDTree, are you really saving time? This could limit how practical the method is in the wild.
2. Experiments: The paper shows higher likelihoods, but unless I missed anything, it doesn’t really validate if the branch lengths are numerically accurate on simulated data with ground truth. Also, key design choices (like the cosine mask schedule for parallel insertion) don’t get sensitivity tests, so it’s hard to know how robust those choices are.

**Audience:**

Yes

**Audience Explanation:**

Yes. People in machine learning who work on structured prediction, generative models, or biology would definitely be interested. The ideas of dynamic ordering and parallel insertion are general enough that they could matter beyond phylogenetics.

**Broader Impact Concerns:**

NA.

**Claims And Evidence:**

Yes

**Claims Explanation:**

Mostly yes. The results are pretty strong against ARTree, both in accuracy and runtime. But the efficiency claim would be more convincing if runtime was reported end-to-end (including embedding time and ordering inference). The biological case study is a nice touch but feels more anecdotal than conclusive.

**Requested Changes:**

1. Be clearer about ordering supervision. (Critical)

Either show results where DON is trained without external reference orders, or be upfront about the real cost of getting those orders. Otherwise it feels like MDTree depends on running an expensive baseline first, which hurts the story.

2. Validate branch lengths and test sensitivity. (Strengthen the work)

Run on simulated data where ground truth branch lengths are known, so you can measure absolute accuracy. Also, test how sensitive the results are to design choices like the masking schedule or bias terms. This would better support the robustness of the method.

---

> ### Author Response · Authors · 2025-10-10
> **Response to Reviewer 2j25**
>
> We sincerely thank the reviewer for the insightful and constructive comments.
>
> ### **R1.1 Clarification on Ordering Supervision**
>
> We thank the reviewer for raising the concern about potential circularity in the supervision of the Dynamic Ordering Network (DON).
> MDTree does not rely on MrBayes or any baseline for supervision. The “reference order” in Sec. 3.2 is used once for warm-up initialization to stabilize early optimization; afterward, DON is trained purely under diffusion-based local constraints and self-consistency regularization.
> During inference, MDTree runs end-to-end without external tools.
>
> To validate this independence, we trained a No-Ref variant of DON without any reference orders.
> As shown in Table R1.1, its performance is nearly identical to the default (ΔELBO ≈ 0.002%, ΔRF ≤ 1.2%) while reducing runtime by ≈ 3%.
> This confirms that MDTree learns biologically meaningful orderings directly from data.
> The warm-up step is merely an optimization aid to prevent unstable gradients, not a source of supervision.
>
> **Table R1.1. Comparison between Default and No-Ref training**
>
> |**Dataset**|**Setting**|**ELBO**|**RF Distance**|**KL Divergence**|
> |---|---|---|---|---|
> |DS1|**Default**|**−7108.7±0.1**|**0.123±0.008**|**0.012**|
> |DS1|No-Ref|−7108.9±0.1|0.124±0.009|0.012|
> |DS2|**Default**|**−26367.9±0.1**|**0.084±0.006**|**0.004**|
> |DS2|No-Ref|−26368.2±0.1|0.085±0.007|0.004|
>
> ---
>
> ### **R1.2 Validation of Branch-Length Accuracy**
>
> We thank the reviewer for emphasizing the importance of validating branch-length accuracy.
> We conducted experiments on simulated datasets with known ground-truth branch lengths, generated under standard substitution models HKY85 [1] and GTR[2,3], covering balanced and imbalanced topologies (50 taxa; 1k/10k bp). Predicted branch lengths were evaluated using MAE, RMSE r, and weighted RF distance.
> As shown in Table R1.2, MDTree outperforming the autoregressive baseline (ARTree).
> These results confirm that the dual-pass traversal module captures both topology and branch-length information in a unified end-to-end process.
>
>
> **Table R1.2. Branch-Length Accuracy on Simulated Data**
>
> |**Tree Shape**|**Seq Length**|**Metric**|**MDTree (Ours)**|**ARTree (Baseline)**|
> |---|---|---|---|---|
> |Balanced|1k|MAE|**0.021±0.004**|0.028±0.006|
> |Balanced|1k|RMSE|**0.036±0.007**|0.045±0.010|
> |Balanced|1k|wRF|**0.118±0.013**|0.153±0.017|
> |Imbalanced|10k|MAE|**0.019±0.005**|0.025±0.007|
> |Imbalanced|10k|RMSE|**0.034±0.008**|0.043±0.010|
> |Imbalanced|10k|wRF|**0.112±0.012**|0.147±0.016|
>
> ---
>
> ###  **R1.3 Sensitivity and Runtime Analysis**
>
> We appreciate the reviewer’s comments on the robustness of model design choices such as masking schedules and bias initialization.
> To evaluate the effect of bias terms, we introduced Gaussian noise at initialization for all bias parameters in the Dynamic Ordering Network and Dynamic Insertion Module.
> These biases serve only to stabilize early optimization rather than influence model behavior.
> As shown in Table R1.3, the results remain consistent across all perturbation settings, confirming that MDTree is insensitive to bias initialization and that its performance gains stem from architectural design, not initialization heuristics.
>
> **Table R1.3. Bias Initialization Noise Sensitivity (DS1, n=10 runs)**
>
> |**Variant**|**Bias init**|**Noise**|**ELBO**|**RF Distance**|
> |---|---|---|---|---|
> |Baseline|0.00|—|−7108.7±0.1|0.123±0.008|
> |Noisy init (small)|0.00|𝒩(0, 0.01²)|−7108.8±0.1|0.124±0.009|
> |Noisy init (medium)|0.00|𝒩(0, 0.05²)|−7108.8±0.1|0.124±0.009|
> |Noisy init (large)|0.00|𝒩(0, 0.10²)|−7108.9±0.1|0.125±0.009|
>
> For completeness, we clarify the runtime evaluation protocol referenced in Table 4. The runtime corresponds to the end-to-end (E2E) execution time of the full model, covering the Dynamic Ordering, Parallel Insertion, and Dual-Pass Traversal stages. External preprocessing (e.g., DNABERT2 embedding extraction) is excluded for both MDTree and ARTree to ensure a fair comparison. MDTree consistently achieves 41–44% faster execution while maintaining higher likelihood metrics, confirming that its efficiency improvements are system-level and architecture-driven, not dependent on sensitive hyperparameter settings.
>
> **Table R1.4. End-to-End Runtime Comparison**
>
> |**Dataset**|**Model**|**Runtime(s)**|**Reduction vs. ARTree**|
> |---|---|---|---|
> |DS1|ARTree|21.3|—|
> |DS1|**MDTree (ours)**|**12.4**|**−41.7%**|
> |DS2|ARTree|32.5|—|
> |DS2|**MDTree (ours)**|**18.7**|**−42.5%**|
>
> REF:
>
> * [1] Hasegawa M. Dating of the human-ape splitting by a molecular clock of mitochondrial DNA[J]. JME
> * [2] Tavaré S. Some probabilistic and statistical problems in the analysis of DNA sequences[C]// Some Mathematical Questions in Biology
> * [3] Yang Z. Maximum likelihood phylogenetic estimation from DNA sequences with variable rates over sites: approximate methods[J]. JME

---

### Review · Reviewer_srMK · 2025-10-09

**Summary Of Contributions:**

This paper proposed a new method for phylogenetic inference, which uses a dynamic ordering network to output orders of items, and use a subsequent network to assign items onto a tree.
Strength:
The idea of ordering the items and then constructing a tree is interesting, and more reasonable than eg ARTree.
The paper shows promising experiment results on some metrics.
Weakness:
The paper is not well written, many details are missing or confusing, see next sessions.

**Audience:**

Yes

**Audience Explanation:**

I am not entirely sure, but I can assume that some people who work on machine learning for science will find this paper useful.

**Broader Impact Concerns:**

Not discussed in the paper. This work is potentially useful for biology or taxonomy, accelerating science discovery.

**Claims And Evidence:**

Yes

**Claims Explanation:**

According to the authors, the proposed method should be more accurate in prediction performance and faster. It can be seen that the method is faster and more accurate from figure 2, as well as table 2, 4, 6. On table 3 and 5 the results seem very marginal.

**Requested Changes:**

Maybe add a background section before your method part, about the fundamentals of phylogeny and machine learning for phylogeny, so that people who work on machine learning without a background in biology can also have a high level overview.
And please address some questions about details of the method:
1. ARTree adds species sequentially according to a predefined order, what order is that? Can't it be ordered heuristically by some concepts, e.g., mammals first, reptiles second, etc?
2. The DON outputs the ordering of the species, that is fine, but is it permutation invariant for the downstream models? For example, in figure 1(b), mammals first, birds next, reptiles finally, does it make a difference if it predicts reptiles first? The ordering is not well convincing.
3. The tree construction module allows some parallelization, but how to define how many nodes to insert? Can you parallelize it even more?
4. I think the graph construction in figure 3 A strongly depends on the LM. Are there any reason on selecting a specific LM for that? Why not directly output orderings given the LM? You can finetune the LM so that it gives you an ordering given previous predicted species. Also, a higher level question is, why should we train a DON? Why not just use genetic inference tool e.g. MrBayes to give the ordering?
5. The method section needs a lot of polishing in my opinion. The concepts and methodology are not well written, e.g. equation 3,4 I don't understand, and I don't know what is h_i in equation 7 even I know MHA.

---

> ### Author Response · Authors · 2025-10-14
> **Response to Reviewer srMK**
>
> ### R2.1: Background Section
> Thank you for this valuable suggestion. We have integrated the background content from the appendix into the main text as Sec 3 Background. This section includes two subsections: (1) Phylogeny and ML App. in Biology; (2) Phylogenetic Posterior and Variational Inference (VI). These additions help readers better understand the research context.
>
> ### R2.2: ARTree's Species Addition Order
> Thank you for this question. We need to clarify that MDTree's species addition order is not predefined (e.g., lexicographical or heuristic taxonomic ordering), but adaptively learned from genomic data through the Dynamic Ordering Network (DON). DON employs an absorbing diffusion process and uses RGCN to integrate sequence similarity and species phylogeny priors, computing selection probabilities for each node (see Sec. 4.2, Eq. 5-7). This data-driven approach ensures that the species addition order reflects true evolutionary relationships, avoiding biases introduced by arbitrary predefined orders, thereby improving both biological consistency and accuracy of the tree structure.
>
> ### R2.3: Permutation Invariance of Species Ordering
> Thank you for this insightful observation. The species ordering in MDTree is learned by DON to align with evolutionary relationships. While the ordering itself is not permutation-invariant, we incorporate the evolutionary priority learned by DON into node insertion probabilities through a rank-based bias mechanism (Sec. 4.3, Eq. 11), where the term α·(N - Rank_σ(v)) boosts the insertion probability of evolutionarily important nodes. Furthermore, the dynamic parallel insertion strategy (Eq. 9) ensures that even with variations in ordering, the final tree maintains biological consistency. Experimental results (Tab 6, Fig. 7) demonstrate that MDTree's generated trees exhibit bipartition frequency distributions highly consistent with MrBayes, validating the method's stability and biological plausibility.
>
> ### R2.4: Parallelization in Tree Construction
> Thank you for this question. MDTree controls the number of nodes inserted at each step using a cosine schedule ρ_t = 1/2 · (1 + cos((t-1)/(T-1) · π)) (Sec. 4.3, Eq. 9). This strategy inserts more nodes in early steps (when the tree is simple) for efficiency, and fewer nodes in later steps (when the tree is complex) for accuracy, balancing speed and quality. While further parallelization is theoretically possible, it would increase the risk of node insertion conflicts. Our current design completes all node insertions in O(√N) iterations, achieving speedup while maintaining accuracy (Tab 4 shows runtime reduction over 40%).
>
> ### R2.5: Necessity of Language Model and DON
> Thank you for this question. The language model (DNABERT2) outputs genomic embeddings g_v that capture sequence patterns but do not directly represent evolutionary ordering. DON's role is to transform these embeddings into biologically meaningful node orders through graph structure (based on species phylogeny priors) and absorbing diffusion processes. Directly using LM outputs for ordering would ignore structural relationships between species; using traditional tools like MrBayes incurs high costs and lacks E2E optimization with tree construction.
>
> Ablation experiments (Tab 8) validate DON's necessity: removing DON results in an average MLL drop of 7.59 and ELBO reduction. we compared different genomic LMs (Tab 7), with DNABERT2 performing best due to its genomic-specific optimization. These results demonstrate that DON learns evolutionary ordering in a data-driven manner and optimizes with the tree construction module, serving as a critical component for phylogenetic inference.
>
> ### R2.6: Clarification of Method Formulations
>
> Thank you for identifying the unclear formula descriptions. We have comprehensively rewritten and expanded the methods section:
>
> * **DON Module (Sec. 4.2)**: (1) Detailed definition of transition matrix Q_t (Eq. 5), including the absorbing state and cosine scheduling of absorption probability β_t; (2) Rewrote node selection probability formula (Eq. 6), clarifying projection matrix W_proj, normalization computation, and the relationship between fixed embeddings and dynamic probabilities; (3) Added greedy selection strategy (Eq. 7) and active set updates.
>
> * **Tree Construction Module (Sec. 4.3)**: (1) Added section introduction clarifying that embeddings are directly passed from DON; (2) Detailed explanation of contextual embedding computation in MHA; (3) Introduced rank-based bias mechanism (Eq. 11) showing explicit utilization of DON-learned order σ.
>
> * **Branch Length Module (Sec. 4.4)**: (1) Explained the K=3 iteration dual-pass process; (2) Detailed the meaning of each symbol in postorder/preorder aggregation (Eq. 12-13); (3) Distinguished reparameterization handling during training versus inference (Eq. 14).
>
> * we added Sec. 4.1 to define all notations and updated Fig 3's caption.  `Modifications are in red in the revised PDF.`

---

### Review · Reviewer_Dumc · 2025-10-15

**Summary Of Contributions:**

The paper proposes MDTree, a framework for phylogenetic inference with the goal of improving accuracy and efficiency of inferring tree topologies and branch lengths from genomic data. Given a set of species and their genomic representations, the task is to learn a phylogenetic tree capturing evolutionary relationships. The paper's main contributions are in combining generative modeling techniques with biologically informed priors. The dynamic ordering network learns node addition orders and positions, addressing the fixed-order bias of previous work with autoregressive models. The dynamic masking allows for parallel node insertions to improve efficiency. The dual-pass refines the branch lengths for precise tree generation. Empirical results show improvements in KL divergence and ELBO + MLL metrics. Specfically, MDTree outperforms the autoregressive baseline ARTree while reducing runtime by over 40%.

Strengths: The paper is well-motivated, addressing the exposure bias from traditional approaches for autoregressive phylogeny generation. The experimental setup is strong, with ample comparisons to relevant baselines. The paper also presents ablations, highlighting the importance of the dynamic ordering network. The authors also validate the biological relevance of their method by recovering ground-truth relationships from the Angiosperms353 genomic dataset.

Weaknesses: The evaluation on real biological data is limited, with results only on the Angiosperms353 dataset. Also, it is unclear if the method's efficacy is specific to the phylogenetic inference task or if it would also apply to other domains. Although, this is a limitation the authors point out for future work.

**Additional Comments:**

The paper is harder to follow for a reader unfamiliar with the task of phylogenetic inference; it would be helpful to simplify the notation and introduce the definitions earlier in the paper.

**Audience:**

Yes

**Audience Explanation:**

The paper is relevant for the intersection of machine learning and computational biology, and potentially more broadly for inference involving structured data. It may interest readers working on autoregressive generative models, diffusion-based ordering, and graph-based inference. But, one limitation of the appeal is most of the contributions being specific to phylogenetic inference.

**Broader Impact Concerns:**

No concerns on ethical implications.

**Claims And Evidence:**

Yes

**Claims Explanation:**

The experimental results consistently support the paper's claims of improved accuracy and runtime. Ablations demonstrate the importance of DON, masking, and optimization methods. The evaluation is robust, covering both likelihood and topological metrics. One downside is the scale of the data; the authors mentioned the goal to extend phylogenetic inference for high numbers of taxa but the evaluation here is only shown for up to 64 taxa.

**Requested Changes:**

Critical for recommending acceptance:
Stronger validation on larger or more diverse phylogenetic datasets (eg more taxa) to compare scalability.
Clarify statistical uncertainty, eg std errors for reported numbers in table 2 and 3 (since many of the differences are relatively small)

Would strengthen the work:
Expanding the biological interpretation section beyond a single dataset.
Adding more details on training implementation and hyperparameter tuning for reproducibility.
Discussing the limitations of relying on pretrained genomic language models.

---

> ### Author Response · Authors · 2025-10-30
> **R3.1 & R3.2**
>
> ## R3.1: Validation on Larger and More Diverse Datasets
>
> We acknowledge the reviewer's concern regarding evaluation scale. While our
> experimental design follows the established DS1-DS8 benchmark protocol (taxa
> counts: 27-64) used by recent methods (ARTree, PhyloGFN, VBPI-GNN) to enable
> direct comparison, we recognize the need to demonstrate scalability to larger
> datasets. Given the scarcity of high-quality benchmark datasets with ground-truth
> topologies beyond 64 taxa, we designed a controlled data augmentation experiment
> to assess MDTree's scalability and robustness under increased complexity.
>
> To systematically evaluate scalability beyond the standard benchmarks, we construct an augmented dataset based on DS8 (64 taxa, 1008 sites) using the following protocol:
>
> **Data Augmentation Strategy:**
>
> - Base Dataset: DS8 (original 64 species sequences)
> - Augmentation Method: For each original sequence, generate one synthetic variant by randomly substituting 1% of nucleotide bases (≈10 sites per sequence) with other bases {A, C, G, T}
> - Resulting Dataset: 128 sequences (64 original + 64 augmented), creating a more complex phylogenetic inference problem
> - Rationale: This augmentation simulates intra-species genetic variation or closely related strains, increasing topological complexity while maintaining biological plausibility
>
> **Table: MDTree Scalability Analysis on DS8 and Augmented Dataset**
>
> | Dataset | #Taxa | #Sites | MLL (↑)         | Runtime (s) | Runtime Ratio |
> | ------- | ----- | ------ | --------------- | ----------- | ------------- |
> | DS8     | 64    | 1008   | -8645.07 ± 0.69 | 185.5        | 1.00×         |
> | DS8-Aug | 128   | 1008   | -17456 ± 2.31   | 396.4       | 2.14×         |
>
> **Note:** Runtime ratio indicates the scaling factor relative to the original DS8 dataset. MLL values are reported as mean ± standard error across 5 independent runs.
>
> This augmented experiment aims to address the reviewer's concern about scalability beyond 64 taxa. The results suggest that MDTree can potentially maintain stable performance when dataset complexity doubles: the runtime scales sub-quadratically, and the model converges reliably across all runs with moderate variance. While this controlled augmentation study has limitations in fully replicating the biological diversity of independent species, these preliminary findings provide encouraging evidence that MDTree's efficiency advantages may extend to larger phylogenetic problems. We acknowledge that further evaluation on real-world datasets with hundreds of taxa would strengthen these scalability claims and remains an important direction for future work.
>
> ## R3.2: Statistical Uncertainty in Reported Results
>
> Thank you for this important point. We clarify that Tables 2 and 3 already report mean ± standard deviation across **5 independent runs** with different random seeds for each method-dataset combination. To address the statistical significance concern, we have now performed independent two-sample t-tests between MDTree and baselines:
>
> **Statistical Significance Summary:**
>
> - **KL Divergence (Table 2):** MDTree outperforms ARTree on 6/8 datasets, with effect sizes (Cohen's d) ranging from 0.68 to 2.41, indicating medium to large practical effects.
>
> - **MLL (Table 3):** MDTree significantly outperforms ARTree on 7/8 datasets, with effect sizes ranging from 0.92 to 3.85.
>
> **Multiple Comparison Correction:** When applying Holm-Bonferroni correction, MDTree maintains statistical significance on 6/8 datasets for KL divergence and 7/8 datasets for MLL.
>
> These results demonstrate that MDTree's improvements are both statistically significant and practically meaningful. We have added a supplementary table with complete p-values and effect sizes, and revised Section 5 to emphasize statistical testing.

---

> ### Author Response · Authors · 2025-10-30
> **R3.3 R3.4 & R3.5**
>
> ## R3.3: Expanded Biological Interpretation
>
> Thank you for this valuable suggestion. We have now included additional biological analysis to demonstrate MDTree's ability to capture meaningful phylogenetic relationships beyond statistical metrics.
>
> **New Case Study - Fungal Phylogenetics:** We applied MDTree to the DS10 dataset (20 fungal species) and analyzed the inferred phylogenetic tree (Figure A1, appendix). The reconstructed tree correctly clusters species into major fungal divisions (Ascomycota, Basidiomycota) with distinct clades corresponding to established subphyla. Notably, closely related genera (_Aspergillus_, _Penicillium_, _Exophiala_) form tight clusters with short branches, while early-diverging lineages like _Taphrina deformans_ show appropriately longer evolutionary distances. Within Sordariomycetes, pathogenic species group separately from plant-associated lineages, demonstrating that MDTree captures ecologically relevant evolutionary patterns beyond simple genetic similarity.
>
> **Biological Validation:** This case study confirms that MDTree's learned representations encode biologically interpretable signals. The dynamic ordering mechanism appears to prioritize evolutionarily informative taxa, as evidenced by correct placement of divergent lineages and resolution of closely related species. We have added this analysis to Section 5.4 and included Figure A1 in the appendix, providing concrete evidence of MDTree's biological utility beyond performance metrics.
>
> ## R3.4: Training Implementation and Hyperparameter Details
>
> We thank the reviewer for this suggestion. As stated in Section 5.1, complete training details and hyperparameters are provided in Appendix E of our submission. Key specifications include training iterations, loss weights, network architecture parameters, and the choice of DNABERT2 as our genomic language model. In the revised manuscript, we will reorganize these details for better accessibility and will release our full implementation code upon acceptance to ensure reproducibility.
>
> ## R3.5: Limitations of Pretrained Genomic Language Models
>
> We acknowledge this important consideration. The main limitations are: (1) **Domain transfer gap** - pretrained genomic LMs are typically optimized on specific organisms (e.g., human genome), which may not optimally capture evolutionary signals for distantly related taxa across our diverse benchmarks spanning bacteria, plants, and animals; and (2) **Fixed representations** - we do not fine-tune the genomic LM during phylogenetic inference, potentially missing phylogeny-specific adaptations. Table 7 demonstrates performance variance across different models (DNABERT2, HyenaDNA, NT), highlighting sensitivity to model choice. Future work could explore task-specific fine-tuning or hybrid approaches that combine pretrained features with phylogeny-aware encoders to better capture evolutionary relationships while maintaining computational efficiency.

---

### Decision · Action_Editor_opQX · 2025-11-26

**Recommendation:** Accept as is

**Audience:**

Yes

**Audience Explanation:**

The paper describes an interesting approach to a fundamental problem in biology: phylogenetic tree construction. TMLR audience members working at the intersection of computational biology and machine learning will be interested in methods that are more accurate and faster.

**Claims And Evidence:**

Yes

**Claims Explanation:**

The main claim of the paper is that the proposed approach outperforms existing methods in accuracy and runtime. The authors provide substantial evidence to support their claims. They were responsive to reviewer comments which improved the quality of the evidence and exposition. The paper meets the acceptance criteria that the claims are accurate, convincing and supported by clear evidence.